# Beyond Penalization: Diffusion-based Out-of-Distribution Detection and Selective Regularization in Offline Reinforcement Learning

**Qingjun Wang**[1], **Hongtu Zhou**[1], **Hang Yu**[1], **Junqiao Zhao**[1,2] *
**Yanping Zhao**[1], **Chen Ye**[1], **Ziqiao Wang**[1], **Guang Chen**[1,3]
[1]School of Computer Science and Technology, Tongji University, Shanghai, China
[2]MOE Key Lab of Embedded System and Service Computing, Tongji University, Shanghai, China
[3]Shanghai Innovation Institute
{2432069, zhouhongtu, 2432034, zhaojunqiao}@tongji.edu.cn
{2534018, yechen, ziqiaowang, guangchen}@tongji.edu.cn

## Abstract

Offline reinforcement learning (RL) faces a critical challenge of overestimating the value of out-of-distribution (OOD) actions. Existing methods mitigate this issue by penalizing unseen samples, yet they fail to accurately identify OOD actions and may suppress beneficial exploration beyond the behavioral support. Although several methods have been proposed to differentiate OOD samples with distinct properties, they typically rely on restrictive assumptions about the data distribution and remain limited in discrimination ability. To address this problem, we propose **DOSER** (**D**iffusion-based **O**OD Detection and **SE**lective **R**egularization), a novel framework that goes beyond uniform penalization. DOSER trains two diffusion models to capture the behavior policy and state distribution, using single-step denoising reconstruction error as a reliable OOD indicator. During policy optimization, it further distinguishes between beneficial and detrimental OOD actions by evaluating predicted transitions, selectively suppressing risky actions while encouraging exploration of high-potential ones. Theoretically, we prove that DOSER is a $\gamma$-contraction and therefore admits a unique fixed point with bounded value estimates. We further provide an asymptotic performance guarantee relative to the optimal policy under model approximation and OOD detection errors. Across extensive offline RL benchmarks, DOSER consistently attains superior performance to prior methods, especially on suboptimal datasets.

## 1 Introduction

Offline reinforcement learning (RL) has emerged as a powerful paradigm for learning policies exclusively from static datasets, eliminating the need for potentially costly or risky online interactions (Levine et al., 2020). This capability renders it particularly appealing for real-world domains where exploration is constrained, such as robotics, healthcare and autonomous systems. However, directly applying standard off-policy RL algorithms to offline dataset pose a fundamental challenge of *distribution shift*. When the learned policy generates actions that deviate substantially from the training data distribution, value functions tend to extrapolate erroneously, leading to severe value overestimation and ultimately catastrophic performance degradation (Fujimoto et al., 2019).

Existing approaches fall into two categories: 1) Policy constraint methods enforce the learned policy remain close to the behavior policy to avoid OOD actions (Kumar et al., 2019; Wu et al., 2019; Fujimoto & Gu, 2021; Kostrikov et al., 2021), typically relying on variational auto-encoders (VAEs) (Kingma & Welling, 2013) for behavior modeling. While effective in principle, these methods struggle to capture the multi-modal nature of real-world behaviors, often collapsing diverse action distributions into suboptimal averaged outputs within low-density regions (Wang et al., 2022). 2) Value regularization methods offer an alternative by learning conservative Q-functions that pe-

---

*Corresponding Author

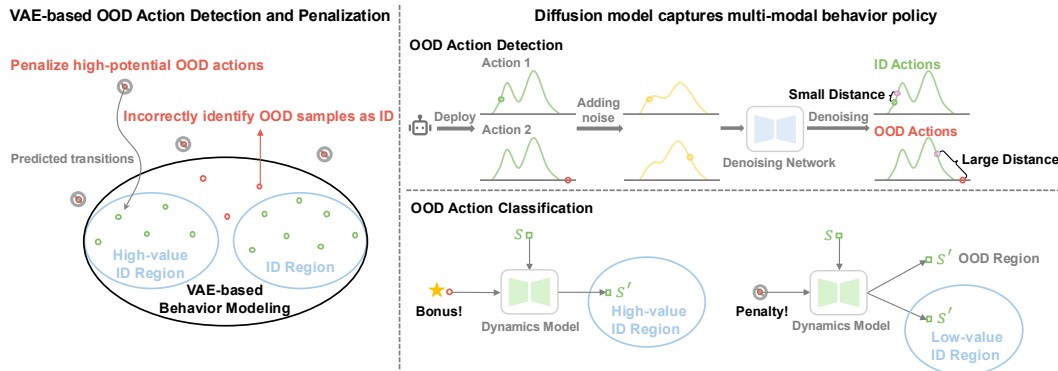

Figure 1: VAE-based behavior modeling methods (left) misidentify OOD actions, whereas uniform penalties suppress high-potential OOD actions. DOSER (right) models multi-modal behavior policy via diffusion model and uses reconstruction error as an OOD indicator, further distinguishing detrimental from beneficial actions for selective regularization.

nalize OOD actions (Kumar et al., 2020; Wu et al., 2021; Bai et al., 2022; Mao et al., 2023). Their effectiveness depends on the underlying OOD identification mechanism, which is a challenging task due to the limited representation capacity of the models used to characterize data distribution. Furthermore, they usually apply uniform penalties across the entire out-of-support region, without considering valuable explorations that could enhance policy performance (Figure 1, left).

Recent efforts have sought to mitigate excessive pessimism by controlling the level of conservatism in a fine-grained manner. CCVL (Hong et al., 2022) conditions the Q-function on a confidence level to learn a spectrum of conservative value estimates, enabling adaptive policies that dynamically adjust pessimism during online evaluation. ACL-QL (Wu et al., 2024) models the behavior policy as a Gaussian distribution and introduces learnable weighting functions to adaptively modulate conservatism at the state-action level. DoRL-VC (Huang et al., 2024) employs a VAE-based detector to separate OOD from ID actions, and further distinguish OOD actions with different properties. Nevertheless, such approaches either rely on Q-ensemble learning to achieve varying degrees of conservatism, incurring additional training overhead, or inherits strong Gaussian assumptions regarding the behavior policy, which fundamentally limit their ability to reliably identify OOD samples.

To address these challenges, we present **DOSER** (**D**iffusion-based **O**OD Detection and **SE**lective **R**egularization), advancing OOD handling through two key innovations (Figure 1, right). First, we utilize diffusion models to achieve OOD detection. By deploying two diffusion models for behavior policy approximation and state distribution modeling, we establish reconstruction errors as theoretically rigorous metrics, avoiding strong Gaussian assumptions while maintaining well-calibrated detection performance. Second, we introduce an adaptive discrimination mechanism that goes beyond binary classification of in-distribution (ID) and OOD. By integrating a learned dynamics model, we distinguish between beneficial OOD actions (those with potential to improve performance while staying within state distribution) and detrimental OOD actions (those likely to induce state distribution shift or value degradation). This fine-grained discrimination enables selective regularization, discouraging hazardous actions while encouraging promising explorations, which yields a robust framework that maintains necessary conservatism while facilitating policy improvement.

The key contributions of this paper are as follows: 1) We propose a diffusion-based approach for OOD detection in offline RL, using reconstruction error as a theoretically grounded metric. 2) We introduce a dual regularization strategy that adaptively adjusts its treatment of OOD actions based on predicted outcomes, suppressing detrimental actions while encouraging beneficial ones. 3) Extensive experiments on D4RL benchmarks demonstrate superior or competitive performance compared to prior methods, with detailed ablations verifying the effectiveness of each component.

## 2 PRELIMINARY

**Offline RL**. We consider the RL problem formulated by the Markov Decision Process (MDP), which is defined as a tuple $(\mathcal{S}, \mathcal{A}, \mathcal{P}, R, \gamma, d_0)$, with state space $\mathcal{S}$, action space $\mathcal{A}$, transition dynamics

$P : \mathcal{S} \times \mathcal{A} \times \mathcal{S} \to [0, 1]$, reward function $R : \mathcal{S} \times \mathcal{A} \to [R_{\min}, R_{\max}]$, discount factor $\gamma \in [0, 1)$, and initial state distribution $d_0 : \mathcal{S} \to [0, 1]$ (Sutton et al., 1998). The goal of RL is to learn a policy $\pi : \mathcal{S} \to \Delta(\mathcal{A})$ that maximizes the expected discounted return $J(\pi) = \mathbb{E}[\sum_{t=0}^{\infty} \gamma^t R(s_t, a_t)]$. For any policy $\pi$, we define the value function as $V^\pi(s) = \mathbb{E}[\sum_{t=0}^{\infty} \gamma^t R(s_t, a_t)|s_0 = s]$, and the Q-function as $Q^\pi(s) = \mathbb{E}[\sum_{t=0}^{\infty} \gamma^t R(s_t, a_t)|s_0 = s, a_0 = a]$. Given that rewards are bounded, the Q-function must lie between $Q_{\min} = R_{\min}/(1-\gamma)$ and $Q_{\max} = R_{\max}/(1-\gamma)$. In offline RL, the agent is limited to learn from a static dataset $\mathcal{D} = \{(s, a, r, s')\}$ collected by a behavior policy $\pi_\beta$, without any interaction with the environment (Lange et al., 2012). We denote the empirical behavior policy as $\hat{\pi}_\beta$, which depicts the conditional action distribution observed in $\mathcal{D}$.

**Diffusion Models**. Diffusion models (Sohl-Dickstein et al., 2015; Ho et al., 2020; Song et al., 2020) have emerged as a powerful class of generative models that excel in capturing complex data distributions. The core idea revolves around a forward diffusion process that gradually perturbs data into noise and a reverse process that learns to reconstruct the original data. Given a clean sample $x_0 \sim p_{\text{data}}(x_0)$ with standard deviation $\sigma_{\text{data}}$, the forward process constructs a sequence of increasingly noisy samples $x_t \sim p(x_t; \sigma_t)$ by adding i.i.d. Gaussian noise with standard deviation $\sigma_t$ that increases along the schedule $\sigma_{\min} = \sigma_0 < \sigma_1 < \cdots < \sigma_N = \sigma_{\max}$. Commonly, $\sigma_{\min}$ is chosen sufficiently small that $p_{\min}(x) \approx p_{\text{data}}(x)$, while $\sigma_{\max}$ is large enough that the final distribution approximates isotropic Gaussian noise, i.e., $p_{\max}(x) \approx \mathcal{N}(x; 0, \sigma_{\max}^2 I)$.

In the original DDPM (Ho et al., 2020) formulation, this process is modeled as a discrete Markov chain. Subsequent works reinterpret it through the lens of stochastic differential equations (SDEs) (Song et al., 2020), describing the evolution of $x_t$ over continuous time $t \in [0, T]$ as:

$$d x_t = f(x_t, t) \, dt + g(t) \, d w_t \tag{1}$$

where $f(\cdot, t)$ and $g(t)$ are the drift and diffusion coefficients, and $w_t$ is a standard Wiener process.

The EDM framework (Karras et al., 2022) refines this paradigm by reparameterizing the diffusion path with differentiable noise schedules $\sigma(t)$. The reverse process is governed by a corresponding probability-flow ODE derived from the forward SDE, which is formulated as:

$$d x_t = -\dot{\sigma}(t)\sigma(t)\nabla_{x_t} \log p_t(x_t) dt \tag{2}$$

where $\dot{\sigma}(t) = \frac{d\sigma}{dt}$ is the time derivative of noise schedule controlling the noise change rate, $\nabla_{x_t} \log p_t(x_t)$ is the score function of the marginal distribution $p_t(x_t)$. The score is approximated by a neural network $\epsilon_\theta(x_t; \sigma_t)$ trained via denoising score matching (Vincent, 2011). The denoising model $\epsilon_\theta$ is trained to predict the true clean sample $x_0$ from its noisy version $x_t = x_0 + \sigma_t \epsilon$ by minimizing the reweighted $L_2$ loss:

$$\mathcal{L}(\theta) = \mathbb{E}_{\sigma_t, x_0 \sim p(x_0), \epsilon \sim \mathcal{N}(0, I)} \left[ \lambda(\sigma_t) || x_0 - \epsilon_\theta(x_t, \sigma_t) ||_2^2 \right], \tag{3}$$

where $\lambda(\sigma_t)$ is the loss weight. Compared to the original DDPM that requires thousands of denoising steps, EDM accelerates sampling by introducing optimized noise schedules and higher-order ODE solvers, achieving high-quality generation within only a few dozen steps.

## 3    DIFFUSION-BASED OOD DETECTION AND SELECTIVE REGULARIZATION

In this section, we present the technical framework of DOSER. We begin by introducing three main components that enable precise detection and classification of OOD actions, then demonstrate the complete integration of these components into a unified algorithmic framework, detailing the practical implementation. Figure 2 provides an overview of the proposed method. For comprehensive theoretical analysis, please refer to Appendix A.

### 3.1    DIFFUSION-BASED BEHAVIOR AND STATE MODELING

The foundation of our approach is to establish two diffusion models that jointly capture the underlying distributions of the offline dataset. We first construct a conditional diffusion model that learns the empirical behavior policy distribution $\hat{\pi}_\beta(a|s)$ by training a denoising network $\epsilon_{\theta_a}(a_t, \sigma_t, s)$ to reconstruct the original action $a_0$ through the following optimization objective:

$$\mathcal{L}(\theta_a) = \mathbb{E}_{\sigma_t, (s, a_0) \sim \mathcal{D}, \epsilon \sim \mathcal{N}(0, I)} \left[ \lambda(\sigma_t) || a_0 - \epsilon_{\theta_a}(a_t, \sigma_t, s) ||_2^2 \right]. \tag{4}$$

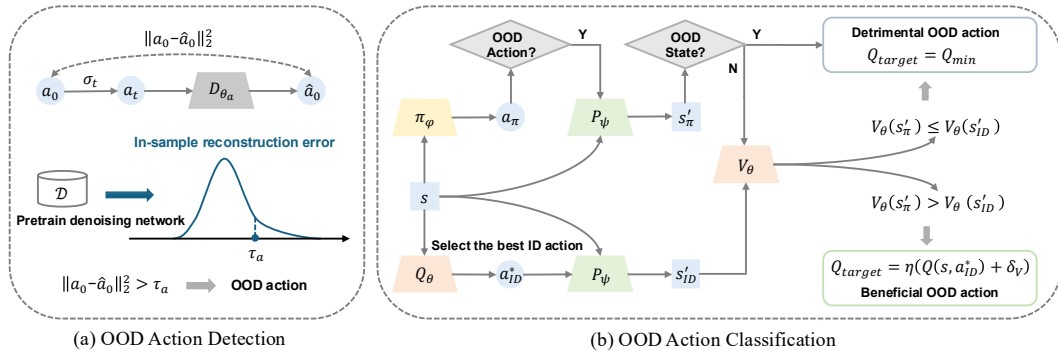

(a) OOD Action Detection                    (b) OOD Action Classification

Figure 2: Overview of the proposed method: (a) Diffusion-based OOD action detection, (b) Integrating the detector to achieve OOD action classification.

where $\boldsymbol{a}_t = \boldsymbol{a}_0 + \sigma_t \boldsymbol{\epsilon}$ is the noisy action with noise scale $\sigma_t$, $\lambda(\sigma_t)$ balances loss scales across noise levels and $\boldsymbol{\epsilon} \sim \mathcal{N}(0, \boldsymbol{I})$.

In parallel, we develop a diffusion model to capture the state distribution $d_0(\boldsymbol{s})$ of the dataset. The corresponding denoising network $\boldsymbol{\epsilon}_{\theta_s}(\boldsymbol{s}_t, \sigma_t)$ is trained to recover the original states $\boldsymbol{s}_0$ from its noisy version $\boldsymbol{s}_t$, using the following reconstruction objective:

$$\mathcal{L}(\theta_s) = \mathbb{E}_{\sigma_t, \boldsymbol{s} \sim \mathcal{D}, \boldsymbol{\epsilon} \sim \mathcal{N}(0, \boldsymbol{I})} \left[ \lambda(\sigma_t) || \boldsymbol{s}_0 - \boldsymbol{\epsilon}_{\theta_s}(\boldsymbol{s}_t, \sigma_t) ||_2^2 \right]. \tag{5}$$

## 3.2 OOD DETECTION VIA RECONSTRUCTION ERROR

Our detection mechanism leverages the denoising capabilities of pretrained diffusion models to identify OOD samples based on reconstruction errors. Given a state-action pair $(\boldsymbol{s}, \boldsymbol{a}_0)$ encountered during policy optimization, we compute its OOD score through a two-step procedure.

First, we sample a noise scale $\sigma_t$ from the training noise schedule and perturb the action as $\boldsymbol{a}_t = \boldsymbol{a}_0 + \sigma_t \boldsymbol{\epsilon}$, where $\boldsymbol{\epsilon} \sim \mathcal{N}(0, \boldsymbol{I})$. The OOD score is then defined as the $L_2$ reconstruction error between the original action and its denoised counterpart:

$$\mathcal{E}_a(\boldsymbol{s}, \boldsymbol{a}_0) = || \boldsymbol{a}_0 - \boldsymbol{\epsilon}_{\theta_a}(\boldsymbol{a}_t, \sigma_t, \boldsymbol{s}) ||_2. \tag{6}$$

Analogously, for state inputs, we measure the reconstruction error between the original state $\boldsymbol{s}_0$ and its denoised version:

$$\mathcal{E}_s(\boldsymbol{s}_0) = || \boldsymbol{s}_0 - \boldsymbol{\epsilon}_{\theta_s}(\boldsymbol{s}_t, \sigma_t) ||_2, \tag{7}$$

where $\boldsymbol{s}_t$ denotes the noise-corrupted state.

Formally, the OOD indicator functions are given by:

$$\mathbb{I}_{\text{ood}}(\boldsymbol{a}_0) = \{\mathcal{E}_a(\boldsymbol{s}, \boldsymbol{a}_0) > \tau_a\}, \quad \mathbb{I}_{\text{ood}}(\boldsymbol{s}_0) = \{\mathcal{E}_s(\boldsymbol{s}_0) > \tau_s\}, \tag{8}$$

where the thresholds $\tau_a$ and $\tau_s$ are set as the $p$-th percentiles of the reconstruction errors on the training dataset $\mathcal{D}$, with $p$ controlling the level of conservatism.

This reconstruction-based method offers three key advantages: 1) Reconstruction error provides a likelihood-free surrogate for distributional alignment, directly measuring conformity to the data manifold without explicit density estimation. 2) Diffusion models naturally capture multi-modal distributions, avoiding the restrictive unimodal Gaussian assumptions of conventional approaches. 3) Detection is efficient, requiring only a single forward pass per sample. Moreover, evaluating errors across multiple randomly sampled diffusion timesteps rather than a fixed noise level improves robustness, since different noise scales correspond to varying levels of information bottleneck in the data distribution.

## 3.3 ADAPTIVE OOD ACTION CLASSIFICATION

Building on the detection framework, we introduce an adaptive classification mechanism to handle OOD actions during policy optimization. Unlike conventional methods that indiscriminately penalize all deviations, our approach distinguishes between *beneficial* and *detrimental* OOD actions through a two-stage assessment process.

---

**Algorithm 1** Diffusion-Based OOD Detection with selective regularization (DOSER)

---

Initialize Q-network $Q_\theta$, V-network $V_\theta$, diffusion behavior model $\epsilon_{\theta_a}$, diffusion state model $\epsilon_{\theta_s}$, policy network $\pi_\phi$, dynamics model $p_\psi$, and target networks $Q_{\theta'}$, $V_{\theta'}$, $\pi_{\phi'}$
**# Model Pretraining**
Pretraining dynamics model $p_\psi$ by minimizing ( 13)
Pretraining diffusion models $\epsilon_{\theta_a}$ and $\epsilon_{\theta_s}$ by minimizing ( 4) and ( 5)
Calculate OOD detection thresholds $\tau_a$ and $\tau_s$ based on in-sample reconstruction error
**for** each iteration **do**
    Sample transition minibatch $\{(\boldsymbol{s}, \boldsymbol{a}, r, \boldsymbol{s}')\}$ from $\mathcal{D}$
    **# Critic Learning**
    Generate action $\boldsymbol{a}_\pi \sim \pi_\phi(\boldsymbol{s})$ and predict the next state $\boldsymbol{s}'_\pi = p_\psi(\boldsymbol{s}, \boldsymbol{a}_\pi)$
    Select the best ID action $\boldsymbol{a}^*_{\mathrm{id}}$ and predict the next state $\boldsymbol{s}'_{\mathrm{id}} = p_\psi(\boldsymbol{s}, \boldsymbol{a}^*_{\mathrm{id}})$
    Calculate the reconstruction errors of policy action and next state by( 6) and( 7)
    Calculate the adaptive bonus $\delta_V = V_\theta(\boldsymbol{s}'_\pi) - V_\theta(\boldsymbol{s}'_{\mathrm{id}})$
    Update $Q_\theta$ and $V_\theta$ by minimizing ( 10) and ( 12)
    **# Actor Learning**
    Update $\pi_\phi$ by minimizing ( 14)
    **# Target Network Update**
    $\theta' \leftarrow \rho\theta + (1 - \rho)\theta'$, $\phi' \leftarrow \rho\phi + (1 - \rho)\phi'$
**end for**

---

For each policy-generated OOD action $\boldsymbol{a}_{\mathrm{ood}}$ in state $\boldsymbol{s}$, we first predict the subsequent state $\boldsymbol{s}'_\pi$ using the learned dynamics model $p_\psi(\boldsymbol{s}'|\boldsymbol{s}, \boldsymbol{a})$, pretrained via supervised learning on the offline dataset $\mathcal{D}$. Since value estimation for OOD states is inherently unreliable, we then evaluate the outcome of $\boldsymbol{a}_{\mathrm{ood}}$ along two dimensions: 1) Whether $\boldsymbol{s}'_\pi$ lies outside the training distribution, determined by the proposed OOD detection mechanism; 2) If $\boldsymbol{s}'_\pi$ is in-distribution, whether $V(\boldsymbol{s}'_\pi)$ exceeds $V(\boldsymbol{s}'_{\mathrm{id}})$, where $\boldsymbol{s}'_{\mathrm{id}}$ denotes the predicted next state after executing the optimal in-distribution action.

Formally, the classification rule for OOD actions is given in Definition 1.

**Definition 1** (Beneficial and detrimental OOD action sets). *Let the beneficial OOD action set $\mathcal{A}^+_{\mathrm{ood}}$ and the detrimental OOD action set $\mathcal{A}^-_{\mathrm{ood}}$ be subsets of the action space $\mathcal{A}$. Then:*

$$\begin{aligned}
\mathcal{A}^+_{\mathrm{ood}} &:= \{\boldsymbol{a} \in \mathcal{A} \mid \mathcal{E}_s(\boldsymbol{s}'_\pi) \leq \tau_s \ \wedge \ V(\boldsymbol{s}'_\pi) \geq V(\boldsymbol{s}'_{\mathrm{id}})\}, \\
\mathcal{A}^-_{\mathrm{ood}} &:= \{\boldsymbol{a} \in \mathcal{A} \mid \mathcal{E}_s(\boldsymbol{s}'_\pi) > \tau_s \ \vee \ V(\boldsymbol{s}'_\pi) < V(\boldsymbol{s}'_{\mathrm{id}})\},
\end{aligned} \tag{9}$$

*where $\boldsymbol{s}'_\pi \sim p_\psi(\cdot|\boldsymbol{s}, \boldsymbol{a}_{\mathrm{ood}})$, $\boldsymbol{s}'_{\mathrm{id}} \sim p_\psi(\cdot|\boldsymbol{s}, \boldsymbol{a}^*_{\mathrm{id}})$, $\boldsymbol{a}^*_{\mathrm{id}} = \arg\max_{\boldsymbol{a} \sim \pi_\beta(\cdot|\boldsymbol{s})} Q(\boldsymbol{s}, \boldsymbol{a})$ is the optimal in-distribution action at state $\boldsymbol{s}$, $\mathcal{E}_s(\cdot)$ is the state reconstruction error defined in (7), and $\tau_s$ is the state OOD threshold.*

Accordingly, detrimental OOD actions are penalized to mitigate overestimation. Conversely, to encourage exploration beyond dataset support, beneficial OOD actions receive an adaptive bonus $\delta_V = V(\boldsymbol{s}'_\pi) - V(\boldsymbol{s}'_{\mathrm{id}})$. This compensates for extrapolation errors in value estimation and guides the policy towards high-value regions, even when Q-value estimates for OOD actions remain imperfect.

Therefore, we minimize the following loss for policy evaluation:

$$\begin{aligned}
\mathcal{L}(\theta) = \ &\mathbb{E}_{(\boldsymbol{s}, \boldsymbol{a}, \boldsymbol{s}') \sim \mathcal{D}} \Big[ \underbrace{\big(Q_\theta(\boldsymbol{s}, \boldsymbol{a}) - \big(R(\boldsymbol{s}, \boldsymbol{a}) + \gamma \mathbb{E}_{\boldsymbol{a}' \sim \pi_\beta(\cdot|\boldsymbol{s})}[Q_{\theta'}(\boldsymbol{s}', \boldsymbol{a}')]\big)\big)^2}_{\text{Standard Bellman error}} \Big] \\
&+ \beta\, \mathbb{E}_{\boldsymbol{s} \sim \mathcal{D}, \, \boldsymbol{a} \sim \pi_\phi(\cdot|\boldsymbol{s})} \Big[ \underbrace{\mathbb{I}(\boldsymbol{a} \in \mathcal{A}^-_{\mathrm{ood}}) \cdot (Q_\theta(\boldsymbol{s}, \boldsymbol{a}) - Q_{\min})^2}_{\text{Penalty for detrimental OOD actions}} \Big] \\
&+ \lambda\, \mathbb{E}_{\boldsymbol{s} \sim \mathcal{D}, \, \boldsymbol{a} \sim \pi_\phi(\cdot|s)} \Big[ \underbrace{\mathbb{I}(\boldsymbol{a} \in \mathcal{A}^+_{\mathrm{ood}}) \cdot (Q_\theta(\boldsymbol{s}, \boldsymbol{a}) - \eta\,(Q_{\theta'}(\boldsymbol{s}, \boldsymbol{a}^*_{\mathrm{id}}) + \delta_V))^2}_{\text{Bonus for beneficial OOD actions}} \Big]
\end{aligned} \tag{10}$$

where $Q_{\theta'}$ is the target Q-network, $Q_{\min} = R_{\min}/(1 - \gamma)$ is the theoretical minimal Q-value of the MDP. In practical implementation, we approximate $\boldsymbol{a}^*_{\mathrm{id}}$ as:

$$\hat{\boldsymbol{a}}^*_{\mathrm{id}} = \arg\max_{\boldsymbol{a}_i \sim \hat{\pi}_\beta(\cdot|\boldsymbol{s})} Q(\boldsymbol{s}, \boldsymbol{a}_i) \quad \text{for} \quad i = 1, \ldots, N \tag{11}$$

with $N = 10$ empirically balancing computational cost and performance across all tasks.

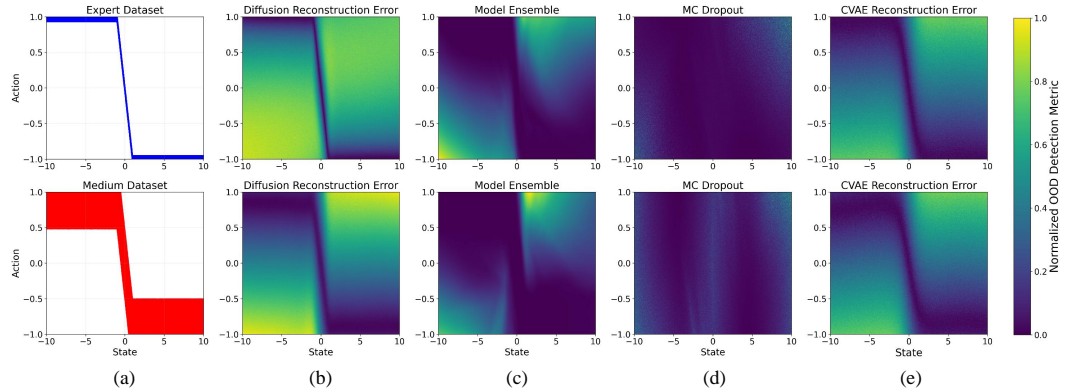

Figure 3: OOD detection experiments on 1D navigation task, where a higher OOD detection metric (reconstruction error or uncertainty estimation) indicates a greater likelihood of being OOD. (a) Two offline datasets with distinct data distributions: expert (top) and medium (bottom). (b) OOD scores across the entire state-action space, evaluated using diffusion-based reconstruction error. (c) OOD scores based on model ensemble uncertainty. (d) OOD scores based on MC dropout uncertainty. (e) OOD scores derived from CVAE-based reconstruction error.

### 3.4 PRACTICAL IMPLEMENTATION

In this section, we provide the practical implementation of our algorithm. [1]

**Value Learning**. Similar to IQL, we perform expectile regression to train the value network.

$$\mathcal{L}(\theta) = \mathbb{E}_{(\boldsymbol{s},r,\boldsymbol{s}')\sim\mathcal{D}} \left[ L_2^\tau(r + \gamma V_{\theta'}(\boldsymbol{s}') - V_\theta(\boldsymbol{s}) \right] \tag{12}$$

where $L_2^\tau(u) = |\tau - \mathbb{I}(u < 0)|u^2$ denotes the asymmetric $L_2$ loss, and $V_{\theta'}$ is the target V-network.

**Dynamics Model**. With the quadruples $(\boldsymbol{s}, \boldsymbol{a}, \boldsymbol{s}')$ in offline dataset $\mathcal{D}$, we train the dynamics model via supervised regression:

$$\mathcal{L}(\psi) = \mathbb{E}_{(\boldsymbol{s},\boldsymbol{a},\boldsymbol{s}')\sim\mathcal{D}}||p_\psi(\cdot|\boldsymbol{s},\boldsymbol{a}) - \boldsymbol{s}'||_2^2 \tag{13}$$

**Policy Learning**. To enhance exploration, we optimize the actor network with maximum entropy regularization:

$$\mathcal{L}(\phi) = \mathbb{E}_{\boldsymbol{s}\sim\mathcal{D},\boldsymbol{a}\sim\pi_\phi(\boldsymbol{s})} \left[ \alpha \log \pi_\phi(\cdot|\boldsymbol{s}) - Q_\theta(\boldsymbol{s},\boldsymbol{a}) \right] \tag{14}$$

where $\alpha$ is dynamically adjusted to maintain target entropy.

**Overall Algorithm**. Putting everything together, we summarize our implementation in Algorithm 1.

## 4 EXPERIMENTS

In this section, we conduct a series of experiments to validate the effectiveness of our proposed method. We aim to answer the following key questions: 1) Is diffusion-based reconstruction error better than existing approaches in detecting OOD samples? 2) How does DOSER perform on standard offline RL benchmarks compared to prior SOTA methods? 3) Does each component in DOSER contribute meaningfully to the overall performance? 4) How sensitive is DOSER to its key hyperparameter? More experimental details and results are provided in Appendix B and C.

### 4.1 OOD DETECTION

To evaluate the effectiveness of diffusion-based reconstruction error for OOD detection, we design a simple 1D navigation task, the discrete state-action space is defined over position $\boldsymbol{s} \in [-10, 10]$ and step size $\boldsymbol{a} \in [-1, 1]$. The reward function is defined as the negative distance to the target state 0, such that rewards increase as the agent approaches the target. By perturbing optimal actions with

---

[1]Code is available at https://github.com/7ingw24/DOSER.

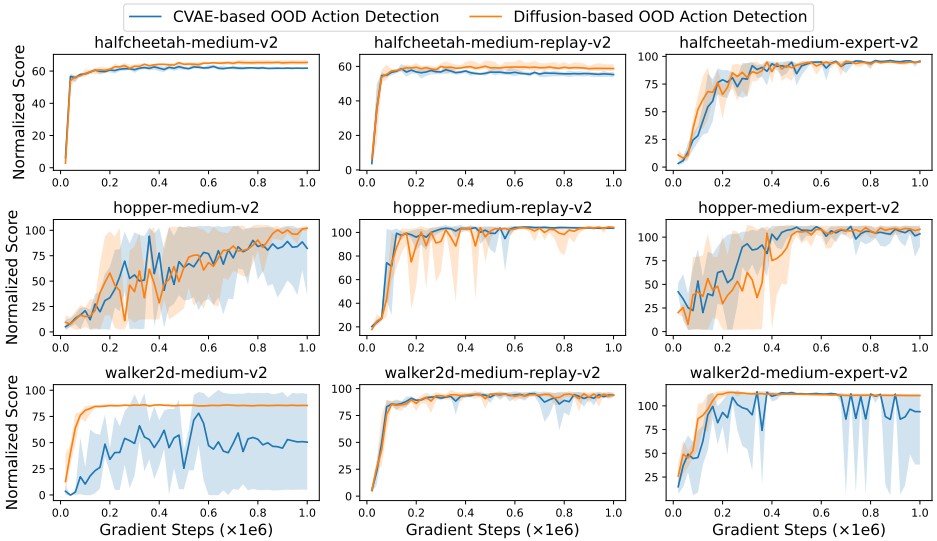

Figure 4: Comparison of OOD action detection performance between CVAE-based reconstruction error and the proposed diffusion-based method in the D4RL MuJoCo domain.

noise of varying scales, we generate two offline datasets, *expert* and *medium*. We then compare our diffusion-based approach against three representative baselines:

**1) Model ensemble.** An ensemble of dynamics models is trained to capture epistemic uncertainty, with OOD samples identified based on high prediction variance across ensemble members.

**2) MC dropout.** Monte Carlo dropout is applied during inference to approximate model uncertainty, where actions with high estimated uncertainty are flagged as OOD.

**3) CVAE-based reconstruction error.** A conditional VAE (CVAE) is trained to model the behavior distribution, and the reconstruction error is used as the OOD indicator.

As shown in Figure 3, our diffusion-based method effectively separates ID and OOD samples across the entire state-action space, whereas baseline methods fail to achieve reliable identification even in this simple setting. In particular, the model ensemble approach frequently misclassifies OOD samples as in-distribution due to its inability to disentangle epistemic and aleatoric uncertainty. Similarly, MC dropout tends to conflate these two sources of uncertainty, while also introducing undesirable stochasticity at inference. Although the CVAE-based reconstruction error baseline shows stronger discrimination than the other two methods, its performance primarily stems from reconstruction ability, while its limited capacity to model multi-modal distributions remains a fundamental limitation (Wang et al., 2022). For more experimental details, please refer to Appendix B.4.

We further compare the OOD action detection performance using CVAE-based reconstruction error with our proposed diffusion-based approach in the D4RL MuJoCo domain (Fu et al., 2020), with results presented in Figure 4. Both methods rely solely on reconstruction error as the detection metric, without incorporating any additional classification or compensation. As illustrated, the CVAE-based method struggles to reliably identify OOD samples in high-dimensional continuous control tasks, which is attributed to its tendency to produce over-smoothed reconstructions, thus diminishing sensitivity to anomalous action inputs. In contrast, our proposed diffusion-based OOD detection consistently delivers superior performance across all evaluated datasets.

## 4.2 COMPARISONS ON D4RL BENCHMARKS

We evaluate the policy performance of DOSER on the standard D4RL benchmark, covering a diverse set of continuous control tasks with varying dataset qualities.

We compare DOSER against a broad range of baselines, including conventional algorithms and SOTA diffusion-based approaches. For policy constraint methods, we include TD3+BC (Fujimoto & Gu, 2021), IQL (Kostrikov et al., 2021) and A2PR (Liu et al., 2024). For value regularization

Table 1: Evaluation results on D4RL benchmark. We report the average normalized scores at the last training iteration over 4 random seeds. Note that m=medium, m-r=medium-replay, m-e=medium-expert. **Bold** indicates the values within 95% of the maximum value.

| Dataset | Conventional methods | | | | | | Diffusion-based methods | | | | | | |
|---|---|---|---|---|---|---|---|---|---|---|---|---|---|
| | TD3+BC | IQL | A2PR | CQL | SVR | ACL-QL | DQL | SfBC | IDQL | QGPO | SRPO | DTQL | DOSER (Ours) |
| halfcheetah-m | 48.3 | 47.4 | **68.6** | 44.0 | 60.5 | **69.8** | 51.5 | 45.9 | 51.0 | 54.1 | 60.4 | 57.9 | **67.5 ± 0.5** |
| hopper-m | 59.3 | 66.3 | **100.8** | 58.5 | **103.5** | 97.9 | 90.5 | 57.1 | 65.4 | 98.0 | 95.5 | **99.6** | **104.0 ± 0.5** |
| walker2d-m | 83.7 | 78.3 | **89.7** | 72.5 | **92.4** | 79.3 | 87.0 | 77.9 | 82.5 | 86.0 | 84.4 | **89.4** | 86.7 ± 1.2 |
| halfcheetah-m-r | 44.6 | 44.2 | 56.6 | 45.5 | 52.5 | 55.9 | 47.8 | 37.1 | 45.9 | 47.6 | 51.4 | 50.9 | **63.0 ± 1.1** |
| hopper-m-r | 60.9 | 94.7 | **101.5** | 95.0 | **103.7** | 99.3 | **101.3** | 86.2 | 92.1 | 96.9 | **101.2** | **100.0** | **104.4 ± 0.6** |
| walker2d-m-r | 81.8 | 73.9 | **94.4** | 77.2 | **95.6** | **96.5** | **95.5** | 65.1 | 85.1 | 84.4 | 84.6 | 88.5 | **94.4 ± 1.3** |
| halfcheetah-m-e | 90.7 | 86.7 | **98.3** | 91.6 | 94.2 | 87.4 | **96.8** | 92.6 | **95.9** | 93.5 | 92.2 | 92.7 | **96.2 ± 0.4** |
| hopper-m-e | 98.0 | 91.5 | **112.1** | 105.4 | **111.2** | 107.2 | **111.1** | **108.6** | **108.6** | 108.0 | 100.1 | **109.3** | **111.5 ± 1.6** |
| walker2d-m-e | **110.1** | **109.6** | **114.6** | 108.8 | **109.3** | **113.4** | **110.1** | **109.8** | **112.7** | **110.7** | **114.0** | **110.0** | **110.9 ± 0.2** |
| **MuJoCo-v2 Average** | 75.3 | 83.3 | **93.0** | 77.6 | 91.4 | 89.6 | 88.0 | 75.6 | 82.1 | 86.6 | 87.1 | 88.7 | **93.2** |
| pen-human | 54.9 | 71.5 | - | 35.2 | 73.1 | - | 72.8 | - | - | 73.9 | - | 64.1 | **87.8 ± 14.7** |
| pen-cloned | 63.8 | 37.3 | - | 27.2 | 70.2 | - | 57.3 | - | - | 54.2 | - | **81.3** | 79.3 ± 8.9 |
| **Adroit-v1 Average** | 59.4 | 54.4 | - | 31.2 | 71.7 | - | 65.1 | - | - | 64.1 | - | 72.7 | **83.6** |

methods, we compare against CQL (Kumar et al., 2020), SVR (Mao et al., 2023) and ACL-QL (Wu et al., 2024). For diffusion-based methods, we consider approaches that also leverage diffusion models for behavior cloning, such as DQL (Wang et al., 2022), SfBC (Chen et al., 2022), IDQL (Hansen-Estruch et al., 2023), QGPO (Lu et al., 2023), SRPO (Chen et al., 2023) and DTQL (Chen et al., 2024). Baseline performance is taken from original papers or recent literature. Some baselines did not report results on the pen tasks, and key hyperparameters for reproduction are unavailable, so we mark these entries as "-".

As shown in Table 1, DOSER consistently achieves strong performance, outperforming prior methods on both Gym-MuJoCo and Adroit tasks. Its advantage is particularly pronounced in the more challenging "medium" and "medium-replay" settings, where the datasets contain a significant proportion of suboptimal and heterogeneous behaviors. This highlights the effectiveness of our proposed diffusion-based OOD detection mechanism and its ability of selective regularization. While existing diffusion-based baselines already exhibit improved performance over traditional approaches due to their expressive modeling capacity, DOSER further improves upon them by explicitly classifying OOD actions, which allows for more refined value estimation and better policy improvement. Note that methods such as SVR and A2PR also incorporate behavior modeling into their frameworks, either for value regularization or policy constraint. Specifically, SVR employs a CVAE to approximate the support of the behavior policy and imposes uniform penalties to actions that fall outside this estimated support. Similar to the motivation of DOSER, A2PR introduces an action discrimination mechanism to guide policy optimization. However, A2PR's discriminator is solely applied to in-distribution actions identified by an enhanced CVAE, thereby restricting policy learning to a potentially inaccurate approximation of the dataset support. In contrast to these CVAE-based approaches, DOSER leverages the expressive power of diffusion models for more accurate OOD detection and employs a selective regularization strategy targeted at OOD actions. This enables the learned policy to extrapolate to high-value regions beyond the offline dataset, ultimately contributing to superior empirical performance.

## 4.3 ABLATION STUDY ON COMPONENTS IN DOSER

To systematically validate the effectiveness of each component in the DOSER framework, we conduct ablation studies on two variants.

**1) DOSER w/o AC and VC**. This variant removes both OOD action classification (AC) and value compensation (VC). It relies solely on diffusion-based reconstruction error to detect OOD actions, applying a uniform penalty without distinguishing between beneficial and detrimental cases. This serves as a direct test of the core capability of diffusion models in OOD detection.

**2) DOSER w/o VC**. Building on the baseline above, this variant further differentiates OOD actions by incorporating both next-state distribution modeling and value estimation. Specifically, it identifies OOD actions that either (i) lead to OOD states or (ii) yield lower value outcomes than op-

Table 2: Components ablation across MuJoCo-v2 tasks.

| Method | halfcheetah | | | hopper | | | walker2d | | |
|---|---|---|---|---|---|---|---|---|---|
| | m | m-r | m-e | m | m-r | m-e | m | m-r | m-e |
| DOSER w/o AC and VC | 65.4 ± 1.1 | 58.8 ± 1.6 | 94.9 ± 0.2 | 102.1 ± 1.7 | 104.2 ± 1.3 | 108.3 ± 2.5 | 85.4 ± 0.4 | 94.1 ± 1.5 | 110.8 ± 0.4 |
| DOSER w/o VC | 67.2 ± 0.9 | 61.9 ± 1.5 | 96.0 ± 0.2 | 99.4 ± 4. | 103.2 ± 1.8 | 111.2 ± 3.2 | 85.8 ± 0.6 | 93.0 ± 1.0 | **111.1 ± 0.5** |
| DOSER | **67.5 ± 0.5** | **63.0 ± 1.1** | **96.2 ± 0.4** | **104.0 ± 0.5** | **104.4 ± 0.6** | **111.5 ± 1.6** | **86.7 ± 1.2** | **94.4 ± 1.3** | 110.9 ± 0.2 |

timal ID actions as detrimental, penalizing only those. All other OOD actions are retained without regularization, enabling a more nuanced treatment of OOD behavior.

We keep all hyperparameters fixed across these variants and evaluate performance on MuJoCo locomotion tasks. Table 2 reports the average normalized scores of DOSER and its two ablated variants across nine datasets. The complete learning curves are provided in Appendix C.9.

The results show that even the baseline variant (**DOSER w/o AC and VC**) already performs competitively with existing SOTA methods, confirming the strong effectiveness of diffusion models for OOD action detection. However, its uniform penalization strategy excessively suppresses potentially beneficial OOD actions, leading to noticeable performance degradation. In contrast, the classification-based variant (**DOSER w/o VC**) alleviates this issue by selectively regularizing only detrimental OOD actions, resulting in smaller performance drops. Overall, these findings strongly validate the effectiveness of DOSER's fine-grained classification and compensation mechanism in better balancing conservatism and exploration during policy optimization.

### 4.4 SENSITIVITY ANALYSIS

We compare different OOD detection thresholds in Figure 5(a), set as the $p$-th percentile of in-distribution reconstruction errors. A smaller threshold implies more samples will be identified as OOD, which is beneficial for narrow behavior distributions like in the "medium-expert" dataset, where a larger threshold might overlook OOD samples. For more diverse datasets like "medium" and "medium-replay", larger thresholds are preferred to prevent ID samples from being misclassified as OOD.

We also investigate the impact of the penalty coefficient $\beta$, varying it from $10^{-5}$ to 1, as shown in Figure 5(b). Datasets with narrow distributions require a larger $\beta$ to prevent value overestimation, while more diverse datasets benefit from a smaller $\beta$ to avoid suppressing beneficial OOD actions.

An ablation study on the compensation coefficient $\lambda$ in Figure 5(c) shows that DOSER performs well across a wide range of $\lambda$ values on the more diverse datasets. Setting $\lambda = 0.001$ yields stable performance across datasets, while excessively large values can amplify the compensation effect, leading to value overestimation and disrupting the learning process.

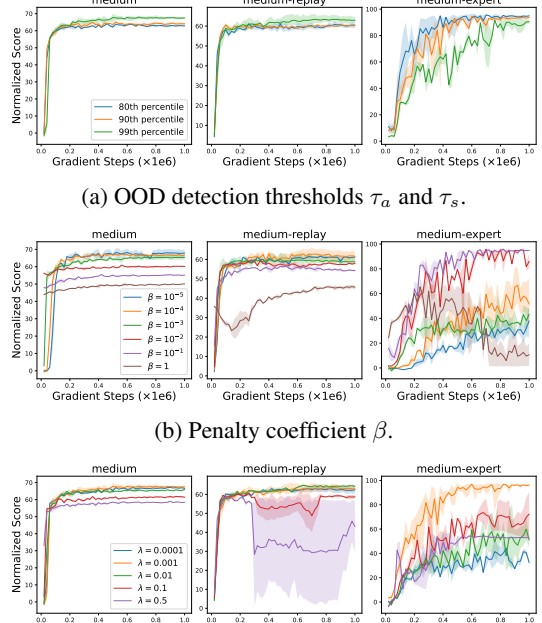

(a) OOD detection thresholds $\tau_a$ and $\tau_s$.

(b) Penalty coefficient $\beta$.

(c) Compensation coefficient $\lambda$.

Figure 5: Ablation study on hyperparameters for halfcheetah tasks.

## 5 RELATED WORKS

**OOD Detection**. Reliable identification of OOD samples is critical for the robustness of machine learning systems. Existing methods primarily fall into two categories: generative-based and reconstruction-based. Generative-based methods leverage probabilistic models to estimate the likelihood of test samples under the learned distribution (Ren et al., 2019), but models such as

Glow (Kingma & Dhariwal, 2018) and VAEs (Kingma & Welling, 2013) often assign higher likelihoods to OOD samples than to ID data (Hendrycks et al., 2018; Nalisnick et al., 2018). Although improvements like likelihood ratios (Ren et al., 2019) and typicality tests (Nalisnick et al., 2019) have been proposed, their reliance on likelihood estimation remains a fundamental limitation. In contrast, reconstruction-based methods (Denouden et al., 2018; Zong et al., 2018) directly measure reconstruction quality, based on the premise that models trained on ID data reconstruct familiar patterns well, while exhibiting significant errors on anomalous inputs. Traditional autoencoders (Lyudchik, 2016) and more recent diffusion-based models (Graham et al., 2023) have shown promising results in this regard, with diffusion models leveraging iterative refinement to further enhance ID reconstruction. Consequently, reconstruction error provides a more reliable signal of distribution shift than likelihood-based metrics, offering improved discriminability between ID and OOD samples.

**OOD Detection in Offline RL**. Offline RL presents additional challenges for OOD detection due to the lack of online interaction. To mitigate the risk of extrapolation error, BCQ (Fujimoto et al., 2019) and SVR (Mao et al., 2023) employ VAEs to approximate the behavior policy, constraining the learned policy to remain within behavior support. However, VAEs often fail to capture multimodal distributions accurately (Wang et al., 2024), resulting in oversimplified generations. Another line of work quantifies uncertainty to identify OOD samples. Model ensemble methods (Lakshminarayanan et al., 2017) identify OOD state-action pairs via predictive variance, with algorithms such as MOPO (Yu et al., 2020) incorporating this uncertainty as a penalty into the reward function. Similarly, Monte Carlo (MC) dropout offers a computationally efficient approximation to Bayesian inference (Gal & Ghahramani, 2016), and has been applied in offline RL for uncertainty-aware OOD detection (Wu et al., 2021). While effective to some extent, both approaches often conflate epistemic and aleatoric uncertainty, which may lead to erroneous identification of OOD actions (Zhang et al., 2023). Alternatively, CQL (Kumar et al., 2020) avoids explicit density estimation by regularizing the Q-function to assign lower values to all unseen actions. This implicit OOD detection eliminates the need for behavior modeling but risks being overly conservative, potentially suppressing valuable actions that lie outside the behavior support but could lead to improved performance.

**Diffusion Models in Offline RL**. Diffusion models have recently emerged as powerful paradigms in RL for modeling multi-modal distributions. This capability is particularly valuable in offline RL settings, where capturing the diversity of behaviors is essential for deriving robust policies. Methods such as Diffusion-QL (Wang et al., 2022) and DAC (Fang et al., 2024) incorporate Q-function guidance into the reverse diffusion process, shaping action generation toward higher-value regions. In contrast, IDQL (Hansen-Estruch et al., 2023) and SfBC (Chen et al., 2022) first pretrain a conditional diffusion model to generate multiple action candidates for a given state, and subsequently resample according to Q-values to select the best action for execution. Notably, while these approaches effectively integrate diffusion models with value functions for policy improvement, their use of diffusion remains largely limited to guiding or selecting actions, none of them fully exploit the inherent properties of diffusion models, such as reconstruction fidelity or noise sensitivity, to directly assess whether state-action pairs lie within the support of the training distribution.

## 6 CONCLUSION

In this work, we proposed DOSER, a framework that mitigates distribution shift through diffusion-based reconstruction error. Unlike prior methods that rely on heuristic uncertainty measures or unreliable likelihood estimates, DOSER leverages the expressive power of diffusion models to compute theoretically grounded reconstruction errors for both behavior policy and state distributions. This provides robust detection metrics that overcome the multi-modality limitations of Gaussian-based approximators. Crucially, DOSER introduces a selective regularization mechanism that classifies OOD samples into beneficial and detrimental actions, enabling suppression of detrimental extrapolations while compensating promising explorations via value-difference bonuses. Extensive experiments demonstrate that DOSER achieves superior or competitive performance compared to state-of-the-art methods, particularly on suboptimal datasets.

Nonetheless, DOSER has two key limitations: 1) its reliance on the accuracy of the diffusion-based reconstruction and the learned dynamics model, and 2) the computational overhead of the iterative diffusion sampling. Future work could focus on enhancing the robustness of dynamics model and improving efficiency via model distillation and accelerated sampling techniques.

## ACKNOWLEDGMENTS

This work is supported by the Shanghai Municipal Science and Shanghai Automotive Industry Science and Technology Development Foundation (No. 2407).

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

APPENDIX

# A THEORETICAL ANALYSIS

In this section, we provide the formal definitions and theoretical analysis in the paper.

## A.1 DEFINITIONS

**Definition 2** (In-sample Bellman operator). *The in-sample Bellman operator is defined as:*

$$\mathcal{T}_{\mathrm{In}}Q(\boldsymbol{s},\boldsymbol{a}) := R(\boldsymbol{s},\boldsymbol{a}) + \gamma\mathbb{E}_{\boldsymbol{s}'\sim P(\cdot|\boldsymbol{s},\boldsymbol{a}),\boldsymbol{a}'\sim\hat{\pi}_\beta(\cdot|\boldsymbol{s}')}\left[Q(\boldsymbol{s}',\boldsymbol{a}')\right], \tag{15}$$

*where $\hat{\pi}_\beta$ is the empirical behavior policy in the dataset.*

Based on Definition 2, DOSER operator is defined as follows.

**Definition 3** (DOSER operator). *From an optimization perspective, (10) lead to the DOSER policy evaluation operator:*

$$\mathcal{T}_{\mathrm{DOSER}}Q(\boldsymbol{s},\boldsymbol{a}) = \begin{cases} \mathcal{T}_{\mathrm{In}}Q(\boldsymbol{s},\boldsymbol{a}) & \text{if } \mathcal{E}_a(\boldsymbol{s},\boldsymbol{a}) \leq \tau_a \\ Q_{\mathrm{adj}}(\boldsymbol{s},\boldsymbol{a}) & \text{otherwise} \end{cases} \tag{16}$$

*where $Q_{\mathrm{adj}}(\boldsymbol{s},\boldsymbol{a})$ is the adjusted Q-target for OOD actions:*

$$Q_{\mathrm{adj}}(\boldsymbol{s},\boldsymbol{a}) = \begin{cases} Q_{\min} & \text{if } \boldsymbol{a} \in \mathcal{A}_{\mathrm{ood}}^- \\ \eta\left(Q(\boldsymbol{s},\boldsymbol{a}_{\mathrm{id}}^*) + \delta_V\right) & \text{if } \boldsymbol{a} \in \mathcal{A}_{\mathrm{ood}}^+ \end{cases} \tag{17}$$

Therefore, DOSER guarantees that the Q-values of ID actions remain unbiased, while underestimate those detrimental OOD actions. By applying value compensation $\delta_V$ to beneficial OOD actions, it incentivizes exploration toward high-potential state-action pairs.

## A.2 THEOREMS

**Theorem 1** (Contraction mapping property). *For arbitrary Q-functions $Q_1$ and $Q_2$ defined on the whole state-action space $\mathcal{S} \times \mathcal{A}$, the DOSER operator $\mathcal{T}_{\mathrm{DOSER}}$ constitutes a $\gamma$-contraction mapping in the $\mathcal{L}_\infty$ norm:*

$$||\mathcal{T}_{\mathrm{DOSER}}Q_1 - \mathcal{T}_{\mathrm{DOSER}}Q_2||_\infty \leq \gamma||Q_1 - Q_2||_\infty. \tag{18}$$

By the Banach fixed-point theorem (Banach, 1922), repeatedly applying $\mathcal{T}_{\mathrm{DOSER}}$ converges to a unique fixed point from any initial Q-function.

**Theorem 2** (Bounded value estimation). *For any policy $\pi$, let $Q_{\mathrm{DOSER}}^\pi$ denote the unique fixed point of the DOSER operator $\mathcal{T}_{\mathrm{DOSER}}$. Then, $Q_{\mathrm{DOSER}}^\pi$ satisfies the following boundedness property for all $(\boldsymbol{s},\boldsymbol{a})$:*

$$Q_{\min} \leq Q_{\mathrm{DOSER}}^\pi(\boldsymbol{s},\boldsymbol{a}) \leq Q_{\mathrm{In}}^\pi(\boldsymbol{s},\boldsymbol{a}_{\mathrm{id}}^*) + \eta\delta_V, \tag{19}$$

*where $Q_{\mathrm{In}}^\pi(\boldsymbol{s},\boldsymbol{a}_{\mathrm{id}}^*) = \max_{\boldsymbol{a}\sim\pi_\beta(\cdot|\boldsymbol{s})} Q_{\mathrm{In}}^\pi(\boldsymbol{s},\boldsymbol{a})$ is the optimal Q-value by iterating the in-sample Bellman operator $\mathcal{T}_{\mathrm{In}}$.*

Since $Q_{\mathrm{In}}^\pi$ corresponds to the fixed point of the in-sample Bellman operator, it yields reliable value estimates within the data distribution. Therefore, Theorem 2 implies that DOSER incurs controlled value overestimation while enabling exploration to high-value regions via the compensation mechanism. The upper bound tightens as the compensation target weight $\eta$ and value difference $\delta_V$ decrease, since smaller values of these parameters directly constrain the magnitude of the value adjustment for OOD actions, aligning the value estimate closer to the in-distribution baseline $Q_{\mathrm{In}}^\pi$.

By dynamically adjusting how OOD actions are treated based on their predicted outcomes, the proposed selective regularization mechanism balances safety and performance, avoiding the pitfalls of binary classification. Crucially, our method preserves standard Q-learning convergence guarantees while enabling safer exploration beyond the behavior policy support.

**Theorem 3** (Bounded critic deviation). *Let $\pi_{\mathrm{ref}}$ denote the reference policy obtained with the true environment dynamics $P$ and without OOD detection error, with $Q^{\pi_{\mathrm{ref}}}$ being its corresponding action-value function. Let $\widehat{\pi}$ denote the learned policy of DOSER under a dynamics model approximation error $\varepsilon_{\mathrm{dyn}}$ and an OOD detection misclassification probability $\varepsilon_{\mathrm{det}}$. Then, the deviation of the learned critic $\widehat{Q}$ from $Q^{\pi_{\mathrm{ref}}}$ is bounded as follows:*

$$\|\widehat{Q} - Q^{\pi_{\mathrm{ref}}}\|_{\infty} \leq \frac{\gamma}{1-\gamma}\left(Q_{\max}\left(C_1\varepsilon_{\mathrm{dyn}} + C_2\varepsilon_{\mathrm{det}}\right) + \eta\delta_V\right), \tag{20}$$

*where $Q_{\max} = \dfrac{R_{\max}}{1-\gamma}$, and $C_1$, $C_2$ are constants that capture the sensitivity of the policy optimization process to dynamics and detection errors respectively.*

**Theorem 4** (Performance gap of DOSER). *Let $\widehat{\pi}$ be the policy learned by DOSER through iterative application of $\mathcal{T}_{\mathrm{DOSER}}$, and let $\pi^*$ denote the optimal policy. Suppose $\delta_f$ represents the function approximation error. Then the performance gap between $\pi^*$ and $\widehat{\pi}$ satisfies*

$$|J(\pi^*) - J(\widehat{\pi})| \leq \delta_f + \frac{CL_P R_{\max}}{1-\gamma}\left(C_1\varepsilon_{\mathrm{dyn}} + C_2\varepsilon_{\mathrm{det}}\right), \tag{21}$$

*where $C_1$, $C_2$ are positive constants, and $L_P$ is the Lipschitz constant of the environment dynamics.*

Consequently, the performance gap is influenced by three key components: the function approximation error $\delta_f$, the OOD detection error $\varepsilon_{\mathrm{det}}$, and the dynamics model approximation error $\varepsilon_{\mathrm{dyn}}$. In our setting, the diffusion model provides a reliable mechanism for OOD detection, as confirmed by extensive experiments, which keeps $\varepsilon_{\mathrm{det}}$ small. Meanwhile, the learned dynamics model maintains stable predictive performance, ensuring that $\varepsilon_{\mathrm{dyn}}$ remains bounded. Therefore, when the diffusion reconstruction error becomes negligible and the dynamics model is sufficiently well fitted, together with a small $\delta_f$, then the right-hand side of the bound vanishes, implying $J(\widehat{\pi}) \to J(\pi^*)$.

## A.3 PROOFS

### A.3.1 PROOF OF THEOREM 1

*Proof.* Let $\mathcal{T}_{\mathrm{DOSER}}$ denote the DOSER operator acting on bounded Q-functions defined on $\mathcal{S} \times \mathcal{A}$. Assume

- Q-functions lie in the Banach space $(\mathcal{B}, \|\cdot\|_{\infty})$ of bounded real functions on $\mathcal{S} \times \mathcal{A}$ with the sup-norm;

- the compensation coefficient satisfies $0 \leq \eta \leq \gamma < 1$;

- the value-compensation term $\delta_V$ is a scalar that does not depend on the Q-function being evaluated (i.e., it is treated as fixed when comparing two Q-functions; if $\delta_V$ depends on $Q$, then it must be Lipschitz continuous in $Q$ with a sufficiently small Lipschitz constant to preserve contraction; here we assume it is fixed for simplicity and clarity).

Let $Q_1, Q_2 \in \mathcal{B}$ be two arbitrary Q-functions. We will bound $\|\mathcal{T}_{\mathrm{DOSER}}Q_1 - \mathcal{T}_{\mathrm{DOSER}}Q_2\|_{\infty}$ by considering the three types of actions that DOSER treats differently: 1) in-distribution actions, 2) detrimental OOD actions, and 3) beneficial OOD actions.

**1) In-distribution actions.** For any $(s, a)$ with $\mathcal{E}_a(s, a) \leq \tau_a$, DOSER reduces to the in-sample Bellman operator

$$\mathcal{T}_{\mathrm{DOSER}}Q(s, a) = \mathcal{T}_{\mathrm{In}}Q(s, a) = R(s, a) + \gamma\,\mathbb{E}_{s' \sim P(\cdot|s,a),\, a' \sim \hat{\pi}_\beta(\cdot|s')}\left[Q(s', a')\right] \tag{22}$$

Hence, the contraction property follows from standard Bellman operator properties:

$$
\begin{aligned}
&\|\mathcal{T}_{\text{DOSER}}Q_1(\boldsymbol{s},\boldsymbol{a}) - \mathcal{T}_{\text{DOSER}}Q_2(\boldsymbol{s},\boldsymbol{a})\|_\infty \\
&= \|\mathcal{T}_{\text{In}}Q_1(\boldsymbol{s},\boldsymbol{a}) - \mathcal{T}_{\text{In}}Q_2(\boldsymbol{s},\boldsymbol{a})\|_\infty \\
&= \|(R(\boldsymbol{s},\boldsymbol{a}) + \gamma\mathbb{E}_{\boldsymbol{s}',\boldsymbol{a}'}[Q_1(\boldsymbol{s}',\boldsymbol{a}')]) - (R(\boldsymbol{s},\boldsymbol{a}) + \gamma\mathbb{E}_{\boldsymbol{s}',\boldsymbol{a}'}[Q_2(\boldsymbol{s}',\boldsymbol{a}')])\|_\infty \\
&= \gamma\max_{\boldsymbol{s},\boldsymbol{a}}|\mathbb{E}_{\boldsymbol{s}',\boldsymbol{a}'}[Q_1(\boldsymbol{s}',\boldsymbol{a}') - Q_2(\boldsymbol{s}',\boldsymbol{a}')]| \\
&\leq \gamma\max_{\boldsymbol{s},\boldsymbol{a}}\mathbb{E}_{\boldsymbol{s}',\boldsymbol{a}'}|Q_1(\boldsymbol{s}',\boldsymbol{a}') - Q_2(\boldsymbol{s}',\boldsymbol{a}')| \\
&\leq \gamma\max_{\boldsymbol{s},\boldsymbol{a}}\|Q_1 - Q_2\|_\infty \\
&= \gamma\|Q_1 - Q_2\|_\infty
\end{aligned}
\tag{23}
$$

Thus for all in-distribution $(\boldsymbol{s},\boldsymbol{a})$, we have

$$
\|\mathcal{T}_{\text{DOSER}}Q_1 - \mathcal{T}_{\text{DOSER}}Q_2\|_\infty \leq \gamma\|Q_1 - Q_2\|_\infty
\tag{24}
$$

**2) Detrimental OOD actions.** For detrimental OOD actions $\boldsymbol{a} \in \mathcal{A}_{\text{OOD}}^-$, Q-target are set to a constant $Q_{\min}$ (independent of the current Q):

$$
\mathcal{T}_{\text{DOSER}}Q(\boldsymbol{s},\boldsymbol{a}) = Q_{\min}
\tag{25}
$$

The difference vanishes for any Q-functions $Q_1$, $Q_2$:

$$
\|\mathcal{T}_{\text{DOSER}}Q_1(\boldsymbol{s},\boldsymbol{a}) - \mathcal{T}_{\text{DOSER}}Q_2(\boldsymbol{s},\boldsymbol{a})\|_\infty = \|Q_{\min} - Q_{\min}\|_\infty = 0 \leq \gamma\|Q_1 - Q_2\|_\infty
\tag{26}
$$

**3) Beneficial OOD actions.** For beneficial OOD actions $\boldsymbol{a} \in \mathcal{A}_{\text{OOD}}^+$, DOSER applies a value compensation, quantified as the difference between the value of the next state $\boldsymbol{s}'_\pi$ reaching by taking the beneficial OOD action $\boldsymbol{a}$ and $\boldsymbol{s}'_{\text{id}}$ after executing the best ID action $\boldsymbol{a}^*_{\text{id}}$:

$$
\begin{aligned}
\mathcal{T}_{\text{DOSER}}Q(\boldsymbol{s},\boldsymbol{a}) &= \eta\left(Q(\boldsymbol{s},\boldsymbol{a}^*_{\text{id}}) + \delta_V\right) \\
&= \eta\left(\max_{\boldsymbol{a}\sim\hat{\pi}_\beta(\cdot|\boldsymbol{s})} Q(\boldsymbol{s},\boldsymbol{a}) + V(\boldsymbol{s}'_\pi) - V(\boldsymbol{s}'_{\text{id}})\right)
\end{aligned}
\tag{27}
$$

where by assumption $\delta_V$ is treated as a fixed scalar w.r.t. the Q-function comparison. Thus

$$
\begin{aligned}
&\|\mathcal{T}_{\text{DOSER}}Q_1(\boldsymbol{s},\boldsymbol{a}) - \mathcal{T}_{\text{DOSER}}Q_2(\boldsymbol{s},\boldsymbol{a})\|_\infty \\
&= \|\eta\left(Q_1(\boldsymbol{s},\boldsymbol{a}^*_{\text{id},1}) + \delta_V\right) - \eta\left(Q_2(\boldsymbol{s},\boldsymbol{a}^*_{\text{id},2}) + \delta_V\right)\|_\infty \\
&= \eta\max_{\boldsymbol{s},\boldsymbol{a}}|Q_1(\boldsymbol{s},\boldsymbol{a}^*_{\text{id},1}) - Q_2(\boldsymbol{s},\boldsymbol{a}^*_{\text{id},2})| \\
&= \eta\max_{\boldsymbol{s},\boldsymbol{a}}|\max_{\boldsymbol{a}} Q_1(\boldsymbol{s},\boldsymbol{a}) - \max_{\boldsymbol{a}} Q_2(\boldsymbol{s},\boldsymbol{a})| \\
&\leq \eta\max_{\boldsymbol{s},\boldsymbol{a}}\|Q_1 - Q_2\|_\infty \\
&\leq \gamma\|Q_1 - Q_2\|_\infty
\end{aligned}
\tag{28}
$$

Combining all three cases, we have $\|\mathcal{T}_{\text{DOSER}}Q_1 - \mathcal{T}_{\text{DOSER}}Q_2\|_\infty \leq \gamma\|Q_1 - Q_2\|_\infty$.

By the Banach fixed-point theorem (Banach, 1922), $\mathcal{T}_{\text{DOSER}}$ admits a unique fixed point in $\mathcal{B}$ and iterative application of $\mathcal{T}_{\text{DOSER}}$ from any initial Q-function converges to that fixed point at rate at most $\gamma$. $\qquad\square$

### A.3.2 Proof of Theorem 2

*Proof.* Suppose the DOSER operator $\mathcal{T}_{\text{DOSER}}$ admits a unique fixed point $Q^\pi_{\text{DOSER}}$ (Theorem 1). Assume the compensation term $\delta_V$ is a scalar that does not depend on the Q-function being evaluated (if $\delta_V$ is estimated from Q, a Lipschitz assumption on this estimator must be made).

We reason by cases according to DOSER's treatment of actions. By Theorem 1, the fixed point $Q^\pi_{\text{DOSER}}$ exists and for each $(\boldsymbol{s},\boldsymbol{a})$ satisfies

$$
Q^\pi_{\text{DOSER}}(\boldsymbol{s},\boldsymbol{a}) = \begin{cases} \mathcal{T}_{\text{In}}Q^\pi_{\text{DOSER}}(\boldsymbol{s},\boldsymbol{a}) & \text{if } \mathcal{E}_a(\boldsymbol{s},\boldsymbol{a}) \leq \tau_a \quad \text{(in-distribution)} \\ Q_{\min} & \text{if } \boldsymbol{a} \in \mathcal{A}_{\text{OOD}}^- \quad \text{(detrimental OOD)} \\ \eta\left(Q^\pi_{\text{DOSER}}(\boldsymbol{s},\boldsymbol{a}^*_{\text{id}}) + \delta_V\right) & \text{if } \boldsymbol{a} \in \mathcal{A}_{\text{OOD}}^+ \quad \text{(beneficial OOD)} \end{cases}
\tag{29}
$$

We show the two inequalities (lower and upper bounds) by treating each action type.

**1) Lower bound:** $Q^\pi_{\text{DOSER}}(s, a) \geq Q_{\min}$.

- *In-distribution actions.* For $\mathcal{E}_a(s, a) \leq \tau_a$, we have the in-sample Bellman fixed-point relation

$$Q^\pi_{\text{DOSER}}(s, a) = R(s, a) + \gamma \, \mathbb{E}_{s' \sim P(\cdot|s,a), \, a' \sim \hat{\pi}_\beta(\cdot|s')} \big[ Q^\pi_{\text{DOSER}}(s', a') \big] \tag{30}$$

Since $R(s, a) \geq R_{\min}$ and the fact that for all successor pairs $Q^\pi_{\text{DOSER}}(s', a') \geq Q_{\min}$, we obtain

$$Q^\pi_{\text{DOSER}}(s, a) \geq R_{\min} + \gamma Q_{\min} = Q_{\min} \tag{31}$$

- *Detrimental OOD actions.* The operator directly assigns $Q^\pi_{\text{DOSER}}(s, a) = Q_{\min}$ for $a \in \mathcal{A}^-_{\text{OOD}}$, so the lower bound holds with equality.

- *Beneficial OOD actions.* For $a \in \mathcal{A}^+_{\text{OOD}}$, they receive value compensation weighted by $\eta \in [0, 1)$. Given that the fixed-point value $Q^\pi_{\text{DOSER}}(s, a^*_{\text{id}})$ is at least $Q_{\min}$, $\delta_V \geq 0$ is satisfied for beneficial OOD actions, and $Q_{\min} < 0$ is strictly negative, we have:

$$Q^\pi_{\text{DOSER}}(s, a) = \eta \big( Q^\pi_{\text{DOSER}}(s, a^*_{\text{id}}) + \delta_V \big) \geq \eta \, Q_{\min} \geq Q_{\min} \tag{32}$$

Combining the three subcases establishes the global lower bound $Q^\pi_{\text{DOSER}}(s, a) \geq Q_{\min}$ for all state-action pairs $(s, a)$.

**2) Upper bound:** $Q^\pi_{\text{DOSER}}(s, a) \leq Q^\pi_{\text{In}}(s, a^*_{\text{id}}) + \eta \delta_V$. Let $Q^\pi_{\text{In}}$ denote the fixed point of the in-sample Bellman operator $\mathcal{T}_{\text{In}}$, by construction $Q^\pi_{\text{DOSER}}(s, a) = Q^\pi_{\text{In}}(s, a)$ for every ID state-action pair $(s, a)$. This upper bound guarantees that DOSER incurs only limited overestimation.

- *In-distribution actions.* If $\mathcal{E}_a(s, a) \leq \tau_a$, then

$$Q^\pi_{\text{DOSER}}(s, a) = Q^\pi_{\text{In}}(s, a) \leq Q^\pi_{\text{In}}(s, a^*_{\text{id}}) \leq Q^\pi_{\text{In}}(s, a^*_{\text{id}}) + \eta \delta_V \tag{33}$$

since $\eta \delta_V \geq 0$.

- *Detrimental OOD actions.* For $a \in \mathcal{A}^-_{\text{OOD}}$:

$$Q^\pi_{\text{DOSER}}(s, a) = Q_{\min} \leq Q^\pi_{\text{In}}(s, a^*_{\text{id}}) + \eta \delta_V \tag{34}$$

- *Beneficial OOD actions.* For $a \in \mathcal{A}^+_{\text{OOD}}$,

$$Q^\pi_{\text{DOSER}}(s, a) = \eta \big( Q^\pi_{\text{DOSER}}(s, a^*_{\text{id}}) + \delta_V \big) \tag{35}$$

Note that $a^*_{\text{id}}$ is an in-distribution action, hence

$$Q^\pi_{\text{DOSER}}(s, a^*_{\text{id}}) = Q^\pi_{\text{In}}(s, a^*_{\text{id}}). \tag{36}$$

Substituting yields

$$Q^\pi_{\text{DOSER}}(s, a) = \eta \big( Q^\pi_{\text{In}}(s, a^*_{\text{id}}) + \delta_V \big) \leq Q^\pi_{\text{In}}(s, a^*_{\text{id}}) + \eta \delta_V \tag{37}$$

Putting together the three cases yields the desired upper bound $Q^\pi_{\text{DOSER}}(s, a) \leq Q^\pi_{\text{In}}(s, a^*_{\text{id}}) + \eta \delta_V$ for all $(s, a)$.

Combining the lower and upper bounds above, we obtain for any state-action pair $(s, a)$

$$Q_{\min} \leq Q^\pi_{\text{DOSER}}(s, a) \leq Q^\pi_{\text{In}}(s, a^*_{\text{id}}) + \eta \delta_V \tag{38}$$

which shows the fixed-point values are uniformly bounded and that DOSER prevents uncontrolled value overestimation while permitting strategic exploration of beneficial out-of-distribution regions. $\square$

A.3.3  PROOF OF THEOREM 3

We begin by introducing three key assumptions and an auxiliary lemma that will be used in the proof.

**Assumption 1** (Dynamics model error bound). *There exists a constant $\varepsilon_{\mathrm{dyn}} \geq 0$ such that the learned dynamics model $\widehat{P}(\cdot \mid \boldsymbol{s}, \boldsymbol{a})$ is uniformly close to the true transition kernel $P(\cdot \mid \boldsymbol{s}, \boldsymbol{a})$ in the $\ell_1$-norm, satisfying for all $(\boldsymbol{s}, \boldsymbol{a}) \in \mathcal{S} \times \mathcal{A}$:*

$$\|\widehat{P}(\cdot \mid \boldsymbol{s}, \boldsymbol{a}) - P(\cdot \mid \boldsymbol{s}, \boldsymbol{a})\|_1 \leq \varepsilon_{\mathrm{dyn}}. \tag{39}$$

**Assumption 2** (OOD detector error bound). *There exists a constant $\varepsilon_{\mathrm{det}} \geq 0$ such that the misclassification probability of the Out-of-Distribution detector is uniformly bounded:*

$$\Pr[\text{detector misclassifies } (\boldsymbol{s}, \boldsymbol{a})] \leq \varepsilon_{\mathrm{det}} \quad \text{for all } (\boldsymbol{s}, \boldsymbol{a}) \in \mathcal{S} \times \mathcal{A}. \tag{40}$$

**Assumption 3** (Policy deviation bound). *There exist constants $C_1, C_2 > 0$, characterizing the sensitivity of the policy optimization to dynamics model and OOD detection errors respectively, such that for all states $\boldsymbol{s} \in \mathcal{S}$:*

$$\|\widehat{\pi}(\cdot \mid \boldsymbol{s}) - \pi_{\mathrm{ref}}(\cdot \mid \boldsymbol{s})\|_{\mathrm{TV}} \leq C_1 \varepsilon_{\mathrm{dyn}} + C_2 \varepsilon_{\mathrm{det}}. \tag{41}$$

*where $\varepsilon_{\mathrm{dyn}}$ and $\varepsilon_{\mathrm{det}}$ are defined in Assumptions 1 and 2.*

**Lemma 1.** *Let $\mu$ and $\nu$ be two probability distributions over a finite set $\mathcal{X}$, and let $f : \mathcal{X} \to \mathbb{R}$ be a bounded function with $\|f\|_\infty \leq M$. Then,*

$$|\mathbb{E}_{x \sim \mu}[f(x)] - \mathbb{E}_{x \sim \nu}[f(x)]| \leq 2M \cdot \|\mu - \nu\|_{\mathrm{TV}}, \tag{42}$$

*where $\|\mu - \nu\|_{\mathrm{TV}} = \sup_{A \subseteq \mathcal{X}} |\mu(A) - \nu(A)|$ is the total variation distance. For a finite set $\mathcal{X}$, this is equivalent to $\|\mu - \nu\|_{\mathrm{TV}} = \frac{1}{2} \sum_{x \in \mathcal{X}} |\mu(x) - \nu(x)|$.*

*Proof.* The expectation difference can be written as:

$$
\begin{aligned}
|\mathbb{E}_\mu[f] - \mathbb{E}_\nu[f]| &= \left| \sum_{x \in \mathcal{X}} f(x)\mu(x) - \sum_{x \in \mathcal{X}} f(x)\nu(x) \right| \\
&= \left| \sum_{x \in \mathcal{X}} f(x)(\mu(x) - \nu(x)) \right| \\
&\leq \sum_{x \in \mathcal{X}} |f(x)| \cdot |\mu(x) - \nu(x)| \\
&\leq M \sum_{x \in \mathcal{X}} |\mu(x) - \nu(x)| \\
&= 2M \cdot \|\mu - \nu\|_{\mathrm{TV}}.
\end{aligned}
\tag{43}
$$

The last equality follows from the definition of total variation distance. □

Now we start the proof of Theorem 3.

*Proof.* We begin by defining the Bellman operator associated with the reference policy $\pi_{\mathrm{ref}}$:

$$\mathcal{T}_{\mathrm{ref}} Q(\boldsymbol{s}, \boldsymbol{a}) := r(\boldsymbol{s}, \boldsymbol{a}) + \gamma \, \mathbb{E}_{\boldsymbol{s}' \sim P(\cdot \mid \boldsymbol{s}, \boldsymbol{a}), \boldsymbol{a}' \sim \pi_{\mathrm{ref}}(\cdot \mid \boldsymbol{s}')}[Q(\boldsymbol{s}', \boldsymbol{a}')] \tag{44}$$

Denote the fixed point of the reference operator as $Q^{\pi_{\mathrm{ref}}}$, so that

$$Q^{\pi_{\mathrm{ref}}} = \mathcal{T}_{\mathrm{ref}} Q^{\pi_{\mathrm{ref}}} \tag{45}$$

DOSER critic constructs a modified Bellman target due to three factors: (i) dynamics model error, (ii) detector misclassification, and (iii) value adjustment. Accordingly, the DOSER Bellman operator can be expressed as:

$$\mathcal{T}_{\mathrm{DOSER}} Q(\boldsymbol{s}, \boldsymbol{a}) := r(\boldsymbol{s}, \boldsymbol{a}) + \gamma \, \mathbb{E}_{\boldsymbol{s}' \sim \widehat{P}(\cdot \mid \boldsymbol{s}, \boldsymbol{a}), \boldsymbol{a}' \sim \widehat{\pi}(\cdot \mid \boldsymbol{s}')}[(Q(\boldsymbol{s}', \boldsymbol{a}') + b(\boldsymbol{s}', \boldsymbol{a}'))] \tag{46}$$

where $\widehat{\pi}$ differs from $\pi_{\mathrm{ref}}$ due to dynamics model error and OOD detector error, $b$ represents the value adjustment applied to the target Q.

Now we compare the difference between the two operators when applied to $Q^{\pi_{\mathrm{ref}}}$. Define

$$\Delta(\boldsymbol{s}, \boldsymbol{a}) := \left| \mathcal{T}_{\mathrm{DOSER}} Q^{\pi_{\mathrm{ref}}}(\boldsymbol{s}, \boldsymbol{a}) - \mathcal{T}_{\mathrm{ref}} Q^{\pi_{\mathrm{ref}}}(\boldsymbol{s}, \boldsymbol{a}) \right| \tag{47}$$

Substituting the operator definitions yields:

$$\Delta(\boldsymbol{s}, \boldsymbol{a}) = \gamma \left| \mathbb{E}_{\widehat{P}, \widehat{\pi}}[Q^{\pi_{\mathrm{ref}}} + b] - \mathbb{E}_{P, \pi_{\mathrm{ref}}}[Q^{\pi_{\mathrm{ref}}}] \right| \leq \gamma \big( (I) + (II) \big) \tag{48}$$

where the components correspond to

$$(I) := \left| \mathbb{E}_{\widehat{P}, \widehat{\pi}}[Q^{\pi_{\mathrm{ref}}}] - \mathbb{E}_{P, \pi_{\mathrm{ref}}}[Q^{\pi_{\mathrm{ref}}}] \right|, \quad (II) := \left| \mathbb{E}_{\widehat{P}, \widehat{\pi}}[b] \right| \tag{49}$$

**Bound on (I):** We decompose (I) into the dynamics model approximation error and policy distribution bias:

$$\begin{aligned}
(I) &= \left| \mathbb{E}_{\widehat{P}, \widehat{\pi}}[Q^{\pi_{\mathrm{ref}}}] - \mathbb{E}_{P, \widehat{\pi}}[Q^{\pi_{\mathrm{ref}}}] + \mathbb{E}_{P, \widehat{\pi}}[Q^{\pi_{\mathrm{ref}}}] - \mathbb{E}_{P, \pi_{\mathrm{ref}}}[Q^{\pi_{\mathrm{ref}}}] \right| \\
&\leq \left| \mathbb{E}_{\widehat{P}, \widehat{\pi}}[Q^{\pi_{\mathrm{ref}}}] - \mathbb{E}_{P, \widehat{\pi}}[Q^{\pi_{\mathrm{ref}}}] \right| + \left| \mathbb{E}_{P, \widehat{\pi}}[Q^{\pi_{\mathrm{ref}}}] - \mathbb{E}_{P, \pi_{\mathrm{ref}}}[Q^{\pi_{\mathrm{ref}}}] \right|
\end{aligned} \tag{50}$$

For the dynamics model error, consider the function $f(\boldsymbol{s}') = \mathbb{E}_{\boldsymbol{a}' \sim \widehat{\pi}(\cdot | \boldsymbol{s}')}[Q^{\pi_{\mathrm{ref}}}(\boldsymbol{s}', \boldsymbol{a}')]$. Since $|Q^{\pi_{\mathrm{ref}}}| \leq Q_{\max}$, the function is bounded by $\|f\|_\infty \leq Q_{\max}$. Applying Lemma 1 with distributions $\mu = \widehat{P}(\cdot | \boldsymbol{s}, \boldsymbol{a})$ and $\nu = P(\cdot | \boldsymbol{s}, \boldsymbol{a})$ gives:

$$\begin{aligned}
\left| \mathbb{E}_{\widehat{P}, \widehat{\pi}}[Q^{\pi_{\mathrm{ref}}}] - \mathbb{E}_{P, \widehat{\pi}}[Q^{\pi_{\mathrm{ref}}}] \right| &= \left| \mathbb{E}_{\boldsymbol{s}' \sim \widehat{P}}[f(\boldsymbol{s}')] - \mathbb{E}_{\boldsymbol{s}' \sim P}[f(\boldsymbol{s}')] \right| \\
&\leq 2Q_{\max} \cdot \|\widehat{P}(\cdot | \boldsymbol{s}, \boldsymbol{a}) - P(\cdot | \boldsymbol{s}, \boldsymbol{a})\|_{\mathrm{TV}}
\end{aligned} \tag{51}$$

Using the definition of TV distance $\|\mu - \nu\|_{\mathrm{TV}} = \frac{1}{2}\|\mu - \nu\|_1 \leq \frac{\varepsilon_{\mathrm{dyn}}}{2}$ and Assumption 1, we have:

$$\left| \mathbb{E}_{\widehat{P}, \widehat{\pi}}[Q^{\pi_{\mathrm{ref}}}] - \mathbb{E}_{P, \widehat{\pi}}[Q^{\pi_{\mathrm{ref}}}] \right| \leq 2Q_{\max} \cdot \frac{\varepsilon_{\mathrm{dyn}}}{2} = Q_{\max} \varepsilon_{\mathrm{dyn}}. \tag{52}$$

For the policy distribution bias, we apply a similar argument in the action space $\mathcal{A}$:

$$\begin{aligned}
\left| \mathbb{E}_{P, \widehat{\pi}}[Q^{\pi_{\mathrm{ref}}}] - \mathbb{E}_{P, \pi_{\mathrm{ref}}}[Q^{\pi_{\mathrm{ref}}}] \right| &\leq 2Q_{\max} \|\widehat{\pi}(\cdot | \boldsymbol{s}') - \pi_{\mathrm{ref}}(\cdot | \boldsymbol{s}')\|_{\mathrm{TV}} \\
&= 2Q_{\max}(C_1 \varepsilon_{\mathrm{dyn}} + C_2 \varepsilon_{\mathrm{det}})
\end{aligned} \tag{53}$$

Therefore, the combined bound for (I) is:

$$(I) \leq Q_{\max}((1 + 2C_1)\varepsilon_{\mathrm{dyn}} + 2C_2 \varepsilon_{\mathrm{det}}) \tag{54}$$

**Bound on (II):** Given $|b| \leq \eta \delta_V$, it follows directly that:

$$(II) = \left| \mathbb{E}_{\widehat{P}, \widehat{\pi}}[b] \right| \leq \mathbb{E}_{\widehat{P}, \widehat{\pi}}|b| \leq \eta \delta_V \tag{55}$$

Thus, for all $(\boldsymbol{s}, \boldsymbol{a})$,

$$\Delta(\boldsymbol{s}, \boldsymbol{a}) \leq \gamma \big( Q_{\max}((1 + 2C_1)\varepsilon_{\mathrm{dyn}} + 2C_2 \varepsilon_{\mathrm{det}}) + \eta \delta_V \big) \tag{56}$$

Consequently, the operator difference is bounded in the supremum norm by:

$$\|(\mathcal{T}_{\mathrm{DOSER}} - \mathcal{T}_{\mathrm{ref}}) Q^{\pi_{\mathrm{ref}}}\|_\infty \leq \gamma \big( Q_{\max}((1 + 2C_1)\varepsilon_{\mathrm{dyn}} + 2C_2 \varepsilon_{\mathrm{det}}) + \eta \delta_V \big) \tag{57}$$

By Theorem 1 in the main paper, the DOSER critic converges to the fixed point of $\mathcal{T}_{\mathrm{DOSER}}$. Thus:

$$\widehat{Q} = \mathcal{T}_{\mathrm{DOSER}} \widehat{Q}. \tag{58}$$

We now bound the final approximation error:

$$\begin{aligned}
\|\widehat{Q} - Q^{\pi_{\mathrm{ref}}}\|_\infty &= \|\mathcal{T}_{\mathrm{DOSER}} \widehat{Q} - \mathcal{T}_{\mathrm{ref}} Q^{\pi_{\mathrm{ref}}}\|_\infty \\
&\leq \|\mathcal{T}_{\mathrm{DOSER}} \widehat{Q} - \mathcal{T}_{\mathrm{DOSER}} Q^{\pi_{\mathrm{ref}}}\|_\infty + \|(\mathcal{T}_{\mathrm{DOSER}} - \mathcal{T}_{\mathrm{ref}}) Q^{\pi_{\mathrm{ref}}}\|_\infty \\
&\leq \gamma \|\widehat{Q} - Q^{\pi_{\mathrm{ref}}}\|_\infty + \gamma \big( Q_{\max}((1 + 2C_1)\varepsilon_{\mathrm{dyn}} + 2C_2 \varepsilon_{\mathrm{det}}) + \eta \delta_V \big)
\end{aligned} \tag{59}$$

Rearranging terms:

$$(1 - \gamma)\|\widehat{Q} - Q^{\pi_{\mathrm{ref}}}\|_\infty \leq \gamma \big( Q_{\max}((1 + 2C_1)\varepsilon_{\mathrm{dyn}} + 2C_2 \varepsilon_{\mathrm{det}}) + \eta \delta_V \big) \tag{60}$$

Absorbing the constants into $C_1$ and $C_2$ yields the final result:

$$\|\widehat{Q} - Q^{\pi_{\mathrm{ref}}}\|_\infty \leq \frac{\gamma}{1 - \gamma} \big( Q_{\max}(C_1 \varepsilon_{\mathrm{dyn}} + C_2 \varepsilon_{\mathrm{det}}) + \eta \delta_V \big) \tag{61}$$

This completes the proof. $\qquad \square$

### A.3.4 PROOF OF THEOREM 4

We first make several common continuity assumptions about the learned $Q$ function and the transition dynamics $P$, which is frequently employed in the theoretical analysis of RL (Gouk et al., 2021; Dufour & Prieto-Rumeau, 2013).

**Assumption 4** (Lipschitz Q). *For all $s \in \mathcal{S}$ and $a_1, a_2 \in \mathcal{A}$, the learned value function is $L_Q$-Lipschitz, then*

$$\|Q(s, a_1) - Q(s, a_2)\| \leq L_Q \|a_1 - a_2\|. \tag{62}$$

**Assumption 5** (Lipschitz P). *For all $s \in \mathcal{S}$ and $a_1, a_2 \in \mathcal{A}$, the transition dynamics is $L_P$-Lipschitz, then*

$$\|P(\cdot \mid s, a_1) - P(\cdot \mid s, a_2)\| \leq L_P \|a_1 - a_2\|. \tag{63}$$

**Lemma 2.** *Under Assumptions 5, the following inequality holds:*

$$\text{TV}(d^{\pi_1} \| d^{\pi_2}) \leq C L_P \max_s \|\pi_1(s) - \pi_2(s)\|, \tag{64}$$

*where $C$ is a positive constant and $d^\pi$ is the state occupancy under policy $\pi$.*

$$d^\pi(s) = (1 - \gamma) \sum_{t=0}^{\infty} \gamma^t \mathbb{E}_\pi \left[ \mathbb{I}[s_t = s] \right]. \tag{65}$$

*Proof.* Please refer to Lemma 1 in (Xiong et al., 2022) Lemma A.5 in (Ran et al., 2023). □

Now we start the proof of Theorem 4.

*Proof.* The proof proceeds by decomposing the overall performance gap between the optimal policy $\pi^*$ and the learned policy $\widehat{\pi}$ into manageable components, then bounding each term individually. Similar to Theorem 3, let $\pi_{\text{ref}}$ denote the ideal reference policy, then

$$\begin{aligned} |J(\pi^*) - J(\widehat{\pi})| &= |J(\pi^*) - J(\pi_{\text{ref}}) + J(\pi_{\text{ref}}) - J(\widehat{\pi})| \\ &\leq |J(\pi^*) - J(\pi_{\text{ref}})| + |J(\pi_{\text{ref}}) - J(\widehat{\pi})| \end{aligned} \tag{66}$$

The first term captures approximation error due to function approximation, we denote is as $\delta_f$. Under the asymptotic regime where the empirical fitting errors vanish, this term can be arbitrarily small. Hence we focus on the second term, which quantifies the performance gap between the learned policy and the reference policy.

$$\begin{aligned} &|J(\pi_{\text{ref}}) - J(\widehat{\pi})| \\ &= \left| \frac{1}{1 - \gamma} \mathbb{E}_{s \sim d^{\pi_{\text{ref}}}} [r(s)] - \frac{1}{1 - \gamma} \mathbb{E}_{s \sim d^{\widehat{\pi}}} [r(s)] \right| \\ &= \frac{1}{1 - \gamma} \left| \sum_s (d^{\pi_{\text{ref}}}(s) - d^{\widehat{\pi}}(s)) r(s) \right| \\ &\leq \frac{1}{1 - \gamma} \sum_s |d^{\pi_{\text{ref}}}(s) - d^{\widehat{\pi}}(s)| |r(s)| \\ &\leq \frac{R_{\max}}{1 - \gamma} \text{TV}(d^{\pi_{\text{ref}}}(s) \| d^{\widehat{\pi}}(s)) \\ &\leq \frac{C L_P R_{\max}}{1 - \gamma} \max_s \|\pi_{\text{ref}}(s) - \widehat{\pi}(s)\| \\ &\leq \frac{C L_P R_{\max}}{1 - \gamma} (C_1 \varepsilon_{\text{dyn}} + C_2 \varepsilon_{\text{det}}) \end{aligned} \tag{67}$$

Combining both error terms yields the overall performance guarantee:

$$|J(\pi^*) - J(\pi_{\text{ref}}^*)| \leq \delta_f + \frac{C L_P R_{\max}}{1 - \gamma} (C_1 \varepsilon_{\text{dyn}} + C_2 \varepsilon_{\text{det}}) \tag{68}$$

□

## B EXPERIMENTAL DETAILS

### B.1 DIFFUSION MODEL FRAMEWORK

We adopt the EDM framework (Karras et al., 2022) to leverage the advantages of continuous-time diffusion models for offline RL. EDM builds upon the continuous-time formulation derived from diffusion processes, which allows us to use an optimized ODE solver for sampling. This solver adaptively determines the steps along the noise level trajectory, significantly reducing the computational load and accelerating generation speed, while maintaining high sample quality compared to sampling with a fixed discrete schedule.

**Noise schedule**. In the DOSER framework, the noise schedule is a crucial component of the diffusion model, defining how the noise levels vary over time. Following the insights from the EDM paper, the noise schedule $\sigma_t$ is sampled from a log-logistic distribution $\sigma_t \sim \text{log-logistic}(\log \sigma_{\text{data}}, s)$, where $\log \sigma_{\text{data}}$ serves as the shape parameter and $s$ as the scale parameter. Using this schedule, a noisy action $\boldsymbol{a}_t$ is constructed as $\boldsymbol{a}_t = \boldsymbol{a}_0 + \sigma_t \boldsymbol{\epsilon}$, with $\boldsymbol{\epsilon} \sim \mathcal{N}(0, \boldsymbol{I})$. The parameters are configured as follows: $\sigma_{\text{data}} = 0.5$, and the noise schedule is clamped between $\sigma_{\min} = 0.02$ and $\sigma_{\max} = 80$.

**Training loss**. The EDM framework precondition the neural network with a $\sigma_t$-dependent skip connection to improve numerical stability. Specifically, the denoising network for behavior policy modeling is defined as follows:

$$\boldsymbol{\epsilon}_{\theta_a}(\boldsymbol{a}_t, \sigma_t, \boldsymbol{s}) = c_{\text{skip}}(\sigma_t)\boldsymbol{a}_t + c_{\text{out}}(\sigma_t)F_{\theta_a}(c_{\text{in}}(\sigma_t); c_{\text{noise}}(\sigma_t)|\boldsymbol{s}) \tag{69}$$

Similarly, the denoising network for state distribution modeling is defined as:

$$\boldsymbol{\epsilon}_{\theta_s}(\boldsymbol{s}_t, \sigma_t) = c_{\text{skip}}(\sigma_t)\boldsymbol{s}_t + c_{\text{out}}(\sigma_t)F_{\theta_s}(c_{\text{in}}(\sigma_t); c_{\text{noise}}(\sigma_t)) \tag{70}$$

where $F_{\theta_a}$ and $F_{\theta_s}$ are the neural networks to be actually trained, $c_{\text{skip}}(\sigma_t)$ modulates the skip connection, $c_{\text{in}}(\sigma_t)$ and $c_{\text{out}}(\sigma_t)$ scale the input and output magnitudes respectively, and $c_{\text{noise}}(\sigma_t)$ maps noise level $\sigma_t$ into a conditioning input for $F_{\theta_a}$ and $F_{\theta_s}$.

We can equivalently express the loss ( 4) with respect to the raw network output $F_{\theta_a}$ in ( 69):

$$\mathbb{E}_{\sigma_t,\boldsymbol{s},\boldsymbol{a},\boldsymbol{\epsilon}}\left[\lambda(\sigma_t)c_{\text{out}}^2(\sigma_t)||F_{\theta_a}(c_{\text{in}}(\sigma_t)\cdot(\boldsymbol{a}+\boldsymbol{\epsilon}); c_{\text{noise}}(\sigma_t)|\boldsymbol{s}) - \frac{1}{c_{\text{out}}(\sigma_t)}(\boldsymbol{a} - c_{\text{skip}}(\sigma_t)\cdot(\boldsymbol{a}+\boldsymbol{\epsilon}))||^2\right] \tag{71}$$

Similarly, the loss ( 5) can be expressed based on ( 70):

$$\mathbb{E}_{\sigma_t,\boldsymbol{s},\boldsymbol{\epsilon}}\left[\lambda(\sigma_t)c_{\text{out}}^2(\sigma_t)||F_{\theta_s}(c_{\text{in}}(\sigma_t)\cdot(\boldsymbol{s}+\boldsymbol{\epsilon}); c_{\text{noise}}(\sigma_t)) - \frac{1}{c_{\text{out}}(\sigma_t)}(\boldsymbol{s} - c_{\text{skip}}(\sigma_t)\cdot(\boldsymbol{s}+\boldsymbol{\epsilon}))||^2\right] \tag{72}$$

According to the variance normalization principles, we follow the practical implementation of EDM in parameter choice:

$$\begin{cases} c_{\text{skip}}(\sigma_t) & = \sigma_{\text{data}}^2/(\sigma_t^2 + \sigma_{\text{data}}^2) \\ c_{\text{out}}(\sigma_t) & = \sigma_t \cdot \sigma_{\text{data}}/\sqrt{\sigma_t^2 + \sigma_{\text{data}}^2} \\ c_{\text{in}}(\sigma_t) & = 1/(\sigma_t^2 + \sigma_{\text{data}}^2) \\ c_{\text{noise}}(\sigma_t) & = \frac{1}{4}\ln(\sigma_t) \\ \lambda(\sigma_t) & = (\sigma_t^2 + \sigma_{\text{data}}^2)/(\sigma_t \cdot \sigma_{\text{data}})^2 \end{cases} \tag{73}$$

### B.2 NETWORK ARCHITECTURE

**Behavior policy and state distribution modeling**. Following Chen et al. (2024), we implement both our behavior policy and state distribution as MLP-based diffusion models. The denoising network for behavior policy $\boldsymbol{\epsilon}_{\theta_a}(\boldsymbol{a}_t, t, \boldsymbol{s})$ is a conditional diffusion model that predicts actions given a noisy action vector $\boldsymbol{a}_t$, diffusion timestep $t$ (encoded via sinusoidal positional embedding), and state condition $\boldsymbol{s}$. In contrast, the denoising network for state distribution $\boldsymbol{\epsilon}_{\theta_s}(\boldsymbol{s}_t, t)$ is an unconditional

diffusion model that predicts states from a noisy state $s_t$ and timestep embedding. Both models share the same base architecture, which consists of a 4-layer MLP with Mish activations and 256 hidden units per layer. The main difference lies in their input dimensions, the behavior policy network additionally concatenates the state condition $s$, while the state distribution network operates without conditioning.

**Critic Networks**. Following the implementation of SVR (Mao et al., 2023), the critic network comprises four Q-networks and two V-networks, each implemented as a 3-layer MLPs with 256 hidden units per layer and ReLU activation functions.

**Actor Network**. The actor network adopts a Tanh-Gaussian policy structure similar to SAC (Haarnoja et al., 2018). It is implemented as a 3-layer MLP with 256 hidden units and ReLU activations in all hidden layers. The network supports both deterministic and stochastic action sampling, while preserving entropy regularization for effective exploration.

**Dynamics Model**. The dynamics model is implemented as a 3-layer MLPs with 256 hidden units and ReLU activations, which takes concatenated state-action pairs as input and predicts both the next state and reward.

### B.3 HYPERPARAMETERS

Diffusion models and networks share the same hyperparameter settings across all tasks. The detailed configurations are provided in Table 3.

Table 3: Hyperparameters for all tasks.

| Hyperparameter | Value |
|---|---|
| Optimizer | Adam (Adam et al., 2014) |
| Learning rate | 3e-4 |
| Learning rate decay | Cosine (Loshchilov & Hutter, 2016) |
| Batch size | 256 |
| Discounted factor | 0.99 |
| Target update rate | 0.005 |
| Policy update frequency | 2 |
| Target network update frequency | 2 |
| Number of sampled actions | 10 |
| Compensation coefficient $\lambda$ | 0.001 |
| Compensation target weight $\eta$ | 0.9 |

To accommodate varying data distributions across different tasks, we employ task-specific hyperparameters including the penalty coefficient $\beta$ for detrimental OOD action penalty, the OOD detection thresholds $\tau_a$ and $\tau_s$ for actions and states, the expectile regression factor $\tau$, and the lower bound of Q-value $Q_{\min}$, with their specific values for each task configuration detailed in Table 4.

### B.4 EXPERIMENTAL DETAILS ON TOY EXAMPLE

For the 1D navigation task illustrated in Figure 6(a), the state space $[-10, 10]$ represents the agent's current position, while actions correspond to step sizes within $[-1, 1]$. The reward function is the negative distance to the target state 0. Based on this reward function, the ground truth Q-function is calculated and depicted in Figure 6(b). To evaluate the performance of different methods, we generate an *expert* dataset and a *medium* dataset, each containing 500,000 transitions. The expert dataset is constructed by perturbing the optimal action derived from the ground truth Q-value with small noise $\epsilon \sim \mathcal{U}[-0.05, 0.05]$, while the medium dataset is generated by adding larger noise $\epsilon \sim \mathcal{U}[-0.5, 0.5]$. The score network in this toy example is implemented as a 4-layer MLP with Mish activations and 256 hidden units per layer.

For the model ensemble method, we employ 5 independently trained neural networks with identical architectures to quantify predictive uncertainty. Each model is a 3-layer MLP with ReLU activations and 128 hidden units. All models are trained in a supervised manner for 100 epochs using the Adam optimizer with a fixed learning rate of 1e-3. During inference, the ensemble estimates epistemic uncertainty by computing the normalized variance across model predictions.

Table 4: Task-specific hyperparameter settings.

| Task | $\beta$ | $\tau_a$ | $\tau_s$ | $\tau$ | $Q_{\min}$ |
|------|---------|----------|----------|--------|-----------|
| halfcheetah-medium-v2 | 0.001 | 99th | 99th | 0.9 | -366 |
| halfcheetah-medium-replay-v2 | 0.001 | 99th | 99th | 0.9 | -366 |
| halfcheetah-medium-expert-v2 | 0.05 | 80th | 80th | 0.7 | -366 |
| halfcheetah-expert-v2 | 0.05 | 80th | 80th | 0.7 | -366 |
| halfcheetah-random-v2 | 0.001 | 99th | 99th | 0.9 | -366 |
| hopper-medium-v2 | 0.001 | 99th | 99th | 0.9 | -125 |
| hopper-medium-replay-v2 | 0.001 | 99th | 99th | 0.9 | -125 |
| hopper-medium-expert-v2 | 0.05 | 80th | 80th | 0.7 | -125 |
| hopper-expert-v2 | 0.05 | 80th | 80th | 0.7 | -125 |
| hopper-random-v2 | 0.001 | 99th | 99th | 0.9 | -125 |
| walker2d-medium-v2 | 0.001 | 99th | 99th | 0.9 | -471 |
| walker2d-medium-replay-v2 | 0.001 | 99th | 99th | 0.9 | -471 |
| walker2d-medium-expert-v2 | 0.05 | 99th | 99th | 0.7 | -471 |
| walker2d-expert-v2 | 0.05 | 99th | 99th | 0.7 | -471 |
| walker2d-random-v2 | 0.001 | 99th | 99th | 0.9 | -471 |
| pen-cloned-v1 | 1 | 60th | 60th | 0.7 | -715 |
| pen-human-v1 | 20 | 80th | 80th | 0.7 | -715 |

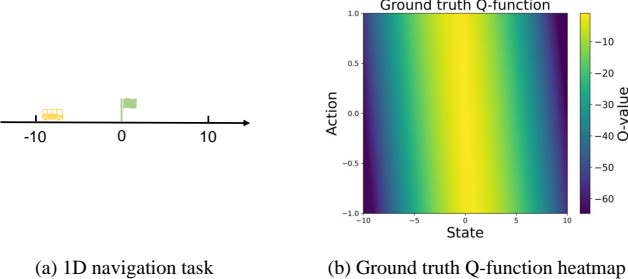

(a) 1D navigation task   (b) Ground truth Q-function heatmap

Figure 6: Toy environment and ground truth Q-function heatmap visualization.

For the MC dropout framework, we adopt a Q-network architecture consisting of 3-layer MLP with 256 hidden units and ReLU activations. Dropout layers with a fixed probability of 0.1 are incorporated to introduce stochasticity during inference. This configuration enables the model to approximate Bayesian inference by maintaining dropout activation during both training and evaluation phases. The Q-network undergoes supervised training for 1,000 epochs using the Adam optimizer with a consistent learning rate of 1e-3. For uncertainty quantification, we performs 20 stochastic forward passes per state-action pair with dropout enabled, computing the epistemic uncertainty as the normalized variance across these Monte Carlo samples.

For the VAE-based method, we adopt a conditional VAE (CVAE) architecture to model the behavior policy distribution and quantify out-of-distribution actions using reconstruction error. The decoder reconstructs the original action through a single output head. The model is trained for 1,000 epochs with the Adam optimizer at a learning rate of 1e-3. During inference, the reconstruction error is computed for state-action pairs by comparing the reconstructed action to the original input action.

### B.5 EXPERIMENTAL DETAILS ON D4RL BENCHMARKS

For all MuJoCo locomotion tasks, we pretrain the diffusion models for both the behavior policy and state distribution for 100,000 gradient steps using the Adam optimizer with a learning rate of 3e-4 and a batchsize of 1024. The dynamics models are also pretrained for 100,000 gradient steps with the same learning rate and batch size. Our algorithm is then trained for 2 million gradient steps to ensure convergence, with policy evaluation performed every 20,000 gradient steps. Results are reported as the average normalized scores over 40 random rollouts, comprising 4 independently trained models and 10 evaluation trajectories per model across all tasks. All experiments are con-

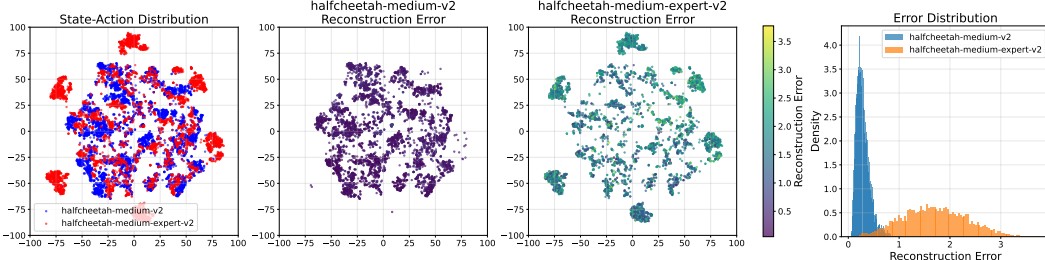

(a) halfcheetah-medium-v2 (ID) vs. halfcheetah-medium-expert-v2 (OOD).

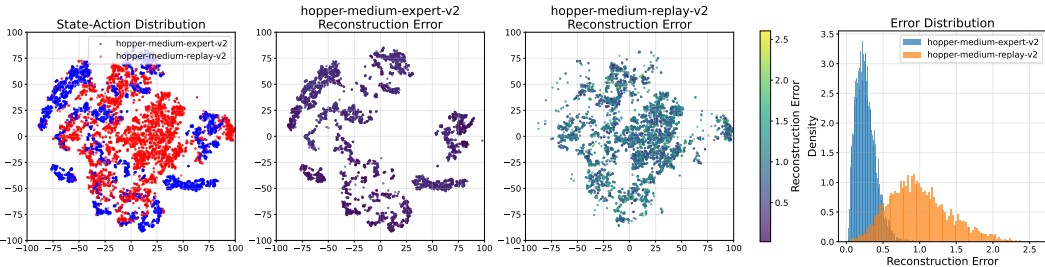

(b) hopper-medium-expert-v2 (ID) vs hopper-medium-replay-v2 (OOD).

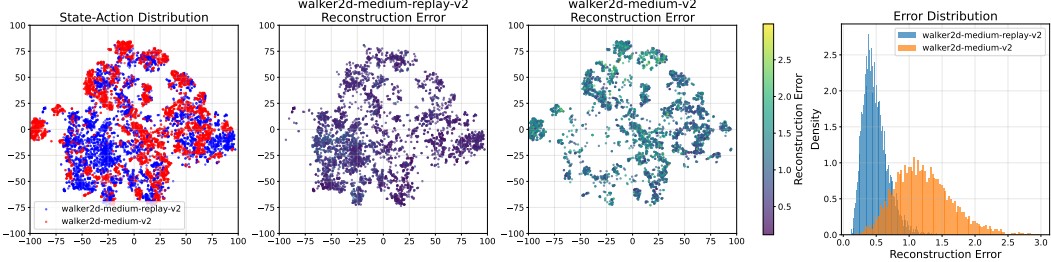

(c) walker2d-medium-replay-v2 (ID) vs walker2d-medium-v2 (OOD).

Figure 7: Diffusion-based reconstruction error distribution across datasets. Diffusion models were trained exclusively on in-distribution (ID) data. From left to right: t-SNE embedding of the state-action distributions; reconstruction errors of ID samples; reconstruction errors of OOD samples; and density plots of error distributions for both ID and OOD samples. The color bar indicates the magnitude of reconstruction error in the second and third columns.

ducted on four NVIDIA GeForce RTX 3090 GPUs, with each experiment taking approximately 30 hours to complete, including both training and evaluation.

## C    ADDITIONAL EXPERIMENTAL RESULTS

### C.1    OOD DETECTION PERFORMANCE ON D4RL BENCHMARKS

To evaluate the ability of our diffusion-based models to distinguish OOD samples, we conduct experiments on the D4RL benchmarks, designating certain datasets as in-distribution (ID) and others as OOD. Specifically, we pretrained diffusion models on the ID datasets, and evaluated their performance on the OOD datasets drawn from the same environment. For each dataset, we randomly sample 5,000 state-action pairs to ensure a balanced comparison. The reconstruction error distributions for the actions are visualized via color-mapped scatter plots and histograms in Figure 7.

Across all environments, OOD datasets consistently exhibit significantly larger reconstruction errors compared to their ID counterparts. This pronounced discrepancy is visually evident in both the color-mapped scatter plots and the histogram plots. In the scatter plots, ID samples are consis-

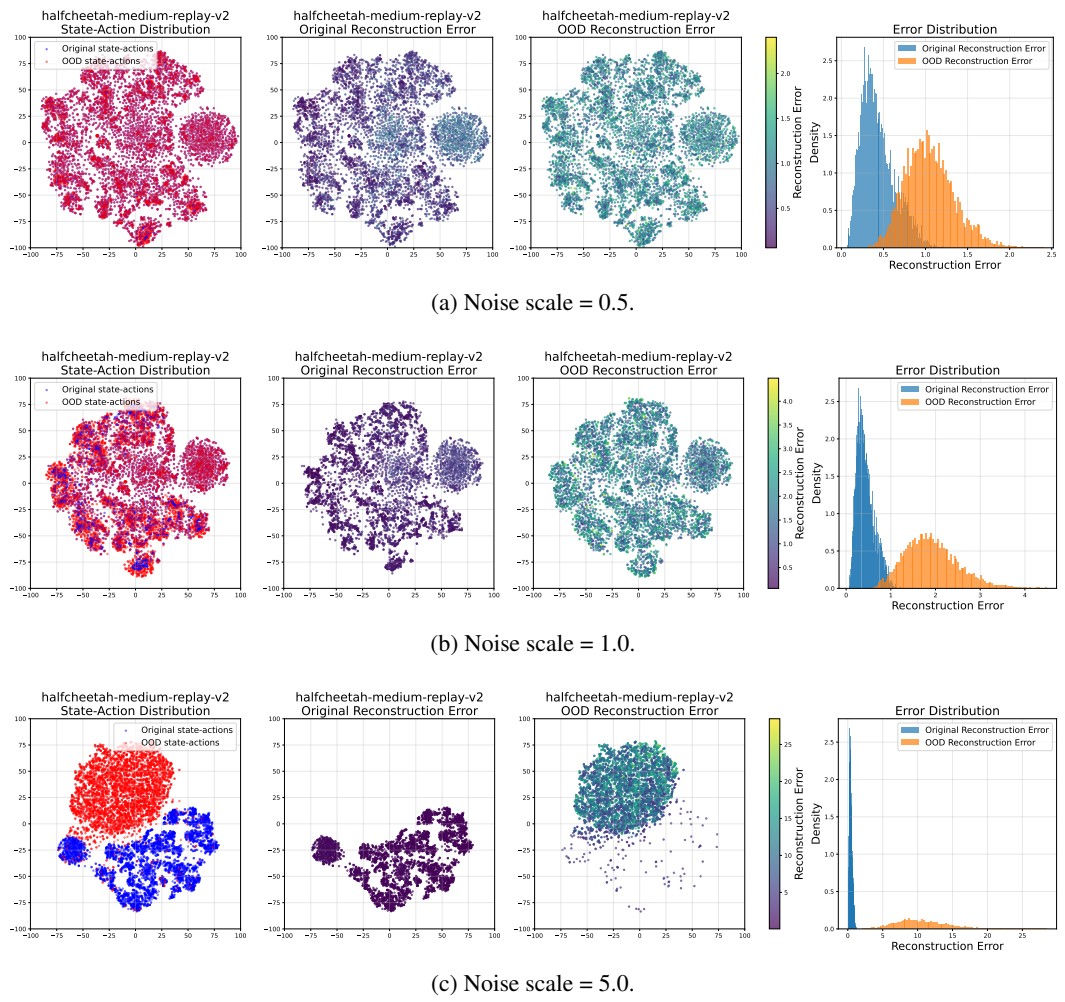

(a) Noise scale = 0.5.

(b) Noise scale = 1.0.

(c) Noise scale = 5.0.

Figure 8: Diffusion-based reconstruction error distributions on original ID datasets and synthetic OOD datasets.

Table 5: OOD detection metrics on synthetic OOD datasets.

| Noise Scale | TP | TN | FP | FN | Accuracy | Precision | Recall | F1-Score | AUROC |
|---|---|---|---|---|---|---|---|---|---|
| 0.5 | 2910 | 4957 | 43 | 2090 | 0.7867 | 0.9854 | 0.5820 | 0.7318 | 0.9637 |
| 1.0 | 4832 | 4957 | 43 | 168 | 0.9789 | 0.9912 | 0.9664 | 0.9876 | 0.9980 |
| 5.0 | 5000 | 4957 | 43 | 0 | 0.9957 | 0.9915 | 1.0000 | 0.9957 | 1.0000 |

tently associated with low reconstruction errors, whereas OOD samples display markedly high error values. Similarly, the histogram plots reveal a distinct shift in the error distributions between ID and OOD samples, with OOD data showing a heavier tail toward higher error values. These results strongly suggest that diffusion-based reconstruction error serves as a robust and effective indicator for OOD detection in this setting.

To provide a more comprehensive quantitative analysis, we construct synthetic OOD datasets as follows. We first sample 5,000 state-action pairs from the original D4RL dataset, and for each pair, we generate a corresponding OOD sample by perturbing the action with standard Gaussian noise using noise scales of 0.5, 1.0, and 5.0, respectively. We evaluate the OOD detection capability of our diffusion-based reconstruction error on these datasets, using the 99-th percentile of reconstruction errors computed from ID samples as the detection threshold. Based on this threshold, we report the counts of true positives (TP), true negatives (TN), false positives (FP), and false negatives (FN), to-

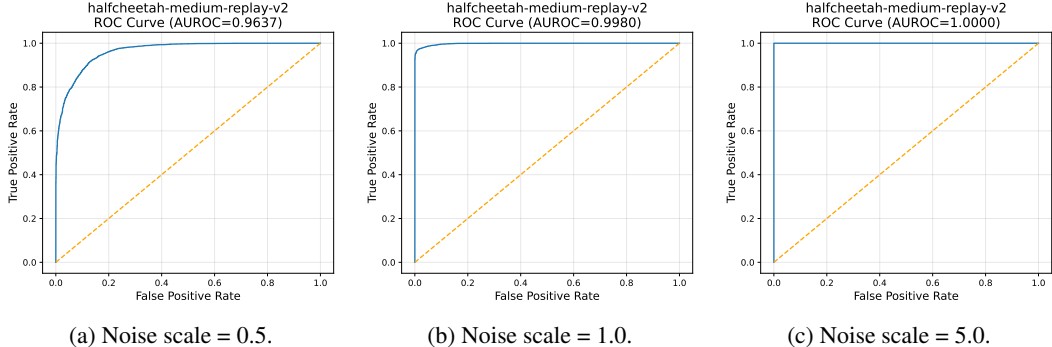

(a) Noise scale = 0.5.  (b) Noise scale = 1.0.  (c) Noise scale = 5.0.

Figure 9: ROC curves for diffusion-based OOD detection under different noise scales.

Table 6: Validation on OOD detection benchmarks.

| Method | KDDCUP | | | KDDCUP-Rev | | | Arrhythmia | | |
|--------|-----------|--------|-------|-----------|--------|-------|-----------|--------|-------|
|        | Precision | Recall | $F_1$ | Precision | Recall | $F_1$ | Precision | Recall | $F_1$ |
| OC-SVM | 0.7457 | 0.8523 | 0.7954 | 0.7148 | **0.9940** | 0.8316 | 0.5397 | 0.4082 | 0.4581 |
| DCN    | 0.7696 | 0.7829 | 0.7762 | 0.2875 | 0.2895 | 0.2885 | 0.3758 | 0.3907 | 0.3815 |
| DSEBM-r | 0.1972 | 0.2001 | 0.1987 | 0.2036 | 0.2036 | 0.2036 | 0.1515 | 0.1513 | 0.1510 |
| DAGMM  | 0.9297 | 0.9442 | 0.9369 | 0.9370 | 0.9390 | 0.9380 | 0.4909 | 0.5078 | 0.4983 |
| GOAD   | - | - | 0.9840 | - | - | **0.9890** | - | - | 0.5200 |
| Ours   | **0.9862** | **0.9937** | **0.9899** | **0.9476** | 0.9144 | 0.9307 | **0.9545** | **0.9545** | **0.9545** |

gether with standard classification metrics including precision, recall, F1-score, and AUROC. These results are summarized in Table 5, and Figure 8 presents the empirical distributions and histograms of reconstruction errors for both ID and OOD samples under different noise scales, while the corresponding ROC curves are shown in Figure 9.

The results show that diffusion-based reconstruction error is highly effective for OOD action detection across different levels of perturbation. When the noise scale is relatively small, the method achieves high precision but moderate recall, indicating that mildly perturbed OOD actions are more difficult to detect. As the noise scale increases, both recall and F1-score improve substantially, reaching nearly perfect detection performance at large perturbations. The AUROC also increases consistently and reaches 1.0 for the largest noise setting, demonstrating that the reconstruction error provides a reliable and discriminative signal for distinguishing ID and OOD actions under challenging distribution shifts.

## C.2 Validation on OOD Detection Benchamrks

To further investigate the effectiveness and generalizability of our proposed diffusion-based OOD detection mechanism beyond the reinforcement learning domain, we conducted additional experiments on three widely used anomaly detection benchmarks: *KDDCUP*, *KDDCUP-Rev*, and *Arrhythmia*. Following the procedure described in the main paper, we compute the OOD score of each sample using the single-step denoising reconstruction error produced by a trained diffusion model. We compare our approach against several state-of-the-art deep learning methods, including OC-SVM (Chen et al., 2001), DCN (Jin et al., 2021), DSEBM-r (Zhai et al., 2016), DAGMM (Zong et al., 2018), and GOAD (Bergman & Hoshen, 2020). Our experimental setup follows GOAD, and the baseline results are taken directly from the respective original publications.

Table 6 summarizes the precision, recall, and F1-scores across all benchmarks. The results demonstrate that our diffusion-based approach consistently achieves high detection accuracy and outperforms or matches existing baselines across all datasets. This robust performance further validates the diffusion model's superior capability in modeling the in-distribution data manifold and confirms the reliability of using the reconstruction error as a general indicator for OOD detection, even across varying data distributions and task contexts.

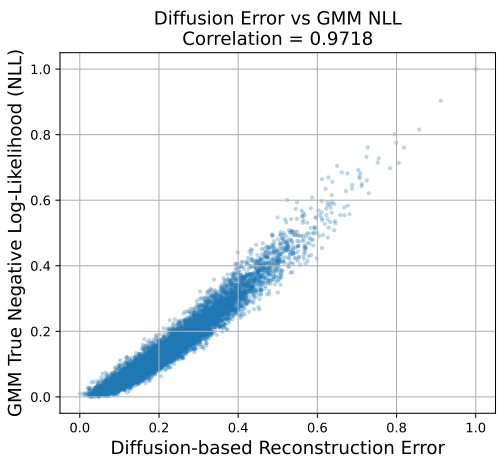 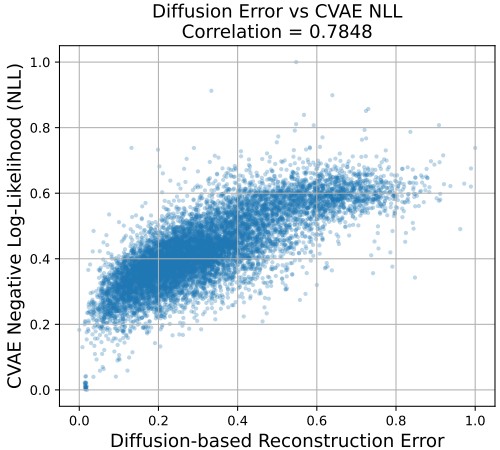

(a) Diffusion-based reconstruction error vs. true NLL on GMM dataset.

(b) Diffusion-based reconstruction error vs. CVAE-based NLL on D4RL dataset.

Figure 10: Correlation analysis between diffusion-based reconstruction error and negative log-likelihood (NLL).

### C.3 QUANTITIVE ANALYSIS BETWEEN DIFFUSION-BASED RECONSTRUCTION ERROR AND NEGATIVE LOG-LIKELIHOOD (NLL)

To further examine whether diffusion-based reconstruction error serves as a meaningful proxy for likelihood estimation, we conduct quantitative correlation analyses on both synthetic and D4RL datasets. Scatter plots illustrating the relationship between diffusion reconstruction error and negative log-likelihood (NLL) are shown in Figure 10.

We first validate the relationship in a Gaussian mixture setting, where the ground-truth density is analytically available. Specifically, we construct a four-component symmetric Gaussian mixture and uniformly sample 10,000 points. Using a diffusion model trained on this distribution, we compute the reconstruction error for each sample and compare it against the true NLL derived from the underlying GMM. The result indicates a very strong Pearson correlation ($\rho = 0.9718$), confirming that the diffusion-based reconstruction error is highly consistent with the true likelihood when the underlying distribution is accurately modeled.

We further evaluate this relationship on the halfcheetah-medium-replay-v2 dataset, where the true behavior policy density is not directly accessible. To approximate the behavior support likelihood, we train a CVAE on the dataset as a reference model and compute its NLL as a surrogate likelihood estimator. The diffusion reconstruction error again exhibits a strong positive correlation with the CVAE-based NLL ($\rho = 0.7848$), indicating that reconstruction error retains a substantial degree of statistical consistency with likelihood even in high-dimensional continuous control environments. However, since CVAEs are known to struggle with accurately modeling multi-modal behavior distributions, the resulting NLL values from the reference model may introduce estimation bias and should therefore be interpreted as only a rough approximation for the true likelihood.

### C.4 ACTION TYPE PROPORTIONS DURING POLICY OPTIMIZATION

To gain a deeper insight into how the action distribution induced by the learned policy evolves over time, we track the proportions ID, beneficial OOD, and detrimental OOD actions throughout the training process. At each training iteration, the statistics are calculated over a sampled batch of size 256. We present the results for three halfcheetah tasks in Figure 11.

Across all datasets, the proportion of ID actions consistently increases as training progresses, accompanied by a corresponding decline in the overall proportion of OOD actions. This trend suggests that the learned policy progressively aligns more closely with the behavioral support during optimization. However, the relative magnitudes of the ID proportions vary considerably across the three datasets.

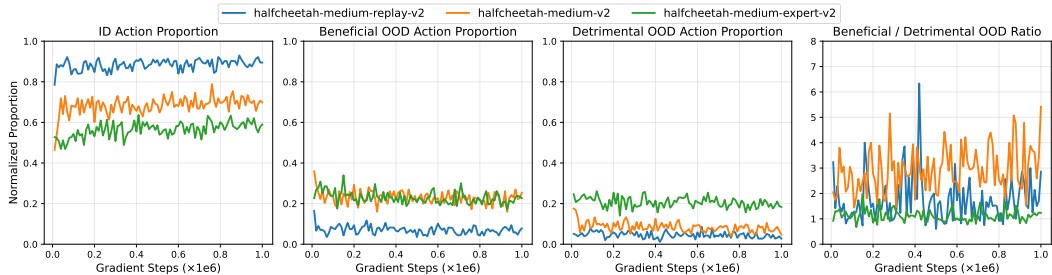

Figure 11: Proportions of different action types during policy optimization.

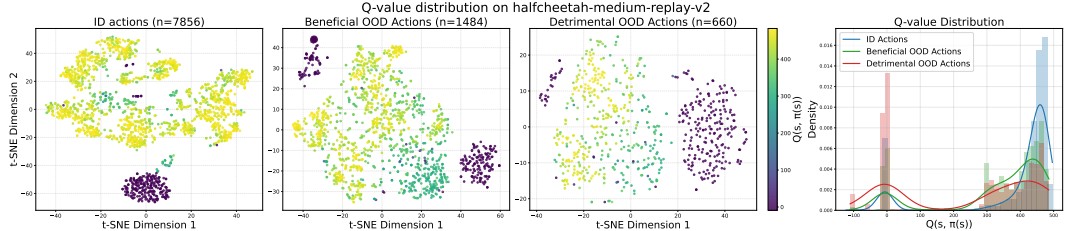

Figure 12: Q-value distributions for different action types.

Specifically, the medium-replay dataset exhibits the largest ID action ratio, followed by the medium dataset, whereas the medium-expert dataset yields the lowest ID proportion. We also visualize the ratio of beneficial to detrimental OOD actions in the rightmost column. For both the medium-replay and medium datasets, the proportion of beneficial OOD actions substantially exceeds that of detrimental ones. Conversely, on the medium-expert dataset, beneficial and detrimental OOD actions appear in almost equal proportion.

These differences can be attributed to the inherent characteristics of the datasets. The medium-expert dataset has a relatively narrow support concentrated around near-optimal trajectories. As a result, the learned policy more frequently generates actions that fall outside this narrow support, leading to a higher overall OOD proportion. Despite this, since the dataset already contains expert demonstrations, the potential performance gain from extrapolation is limited, leading to only a comparable proportion of beneficial and detrimental OOD actions. In contrast, the medium-replay and medium datasets exhibit more diverse distributions of generally suboptimal behaviors. This broader support enables the learned policy to benefit from moderate extrapolation, where slight deviations outside the data manifold can lead to meaningful performance improvements, which is consistent with our empirical results.

## C.5  VISUALIZATION OF Q-VALUE DISTRIBUTION

To investigate whether beneficial OOD actions indeed lead the policy toward higher-value regions, we visualize the learned Q-value landscape. Specifically, we randomly sample 10,000 states from the offline dataset. For each evaluation state, we generate an action using the learned policy and categorize it as an ID action, a beneficial OOD action, or a detrimental OOD action based on the diffusion reconstruction error. We then apply t-SNE to embed the corresponding state-action pairs into a two-dimensional space, where each point is colored according to its Q-value estimate. In addition, we plot the Q-value distributions for the three categories to enable a direct statistical comparison.

As illustrated in Figure 12, beneficial OOD actions exhibit a clearly right-shifted Q-value distribution, indicating that the actions identified as beneficial by our method correspond to regions where the critic consistently predicts higher returns. In contrast, detrimental OOD actions predominantly occupy the low-value region, with their Q-value distribution concentrated near the lower tail. The critic assigns persistently low values to these actions, implying that they are unlikely to yield performance gains and should therefore be suppressed during policy improvement.

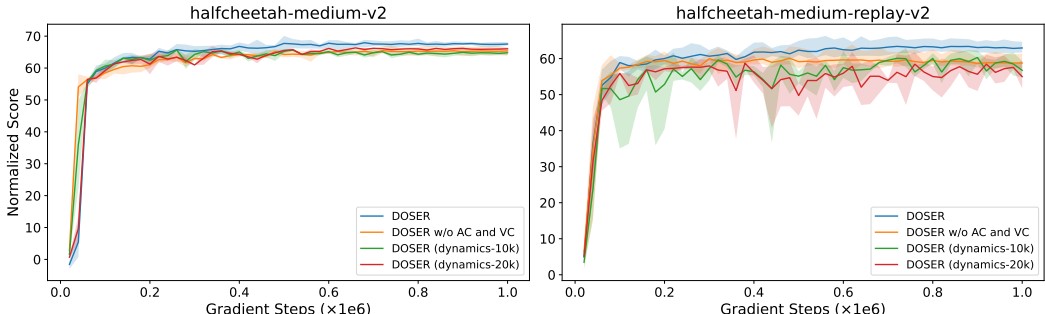

Figure 13: Sensitivity analysis of the dynamics model error.

## C.6  ADDITIONAL SENSITIVITY ANALYSIS

### C.6.1  DYNAMICS MODEL ERROR

To evaluate how the accuracy of the learned dynamics model influences the performance of OOD action classification, we conduct an additional ablation study. Specifically, we pretrain the dynamics model for 100k gradient steps and save the intermediate checkpoints at 10k and 20k steps. These early-stage models exhibit substantially higher prediction error compared to the final checkpoint, providing a controlled mechanism to examine the impact of model inaccuracies. In the subsequent experiments, we replace the fully trained dynamics model with the selected checkpoint while keeping all other components unchanged. We evaluate these variants on two halfcheetah datasets, with the corresponding training curves illustrated in Figure 13.

Our results indicate that employing dynamics models derived from the early checkpoints consistently deteriorates policy performance relative to the fully trained model. Furthermore, on the halfcheetah-medium-replay-v2 dataset, we observe that when the dynamics model is poorly trained, the performance may fall below that of the *DOSER w/o AC and VC* variant introduced in Section 4.3, which intentionally excludes dynamics modeling. This finding highlights that if the dynamics model fails to produce reliable next-state predictions, the resulting misclassification of OOD actions can be more detrimental to overall performance than omitting OOD action classification entirely.

In addition, the observed performance gap between the early-stage and fully trained dynamics models provides further evidence regarding the model's ability to generalize beyond the dataset support. This indicates that the final checkpoint captures meaningful structural regularities of the environment rather than merely memorizing in-distribution transitions. As a result, a well-trained dynamics model can provide sufficiently reliable predictions for moderate OOD actions, which is crucial for the selective regularization mechanism of DOSER.

### C.6.2  THE NUMBER OF CRITIC NETWORKS

In our main experiments, we employ four critic networks for Q-function learning. This design choice follows the implementation of SVR, upon which our training pipeline is partially built. The use of multiple critics has been shown to reduce overestimation bias and stabilize value learning. To examine whether this choice confers any unintended advantage, we perform an additional ablation in which DOSER is trained with only two critic networks while keeping all other components and hyperparameters fixed. We evaluate both settings on the halfcheetah-medium-v2 and halfcheetah-medium-replay-v2 datasets, the corresponding learning curves are presented in Figure 14.

Empirically, we observe that using two critics achieves comparable final performance to the four-critic setting across both tasks, with only a slight difference within an acceptable range. This indicates that DOSER does not rely on the increased critic ensemble size to obtain its performance gains. While additional critics can enhance robustness during training, the core algorithmic contributions of DOSER remain effective under the standard two-critic setup commonly used in offline RL.

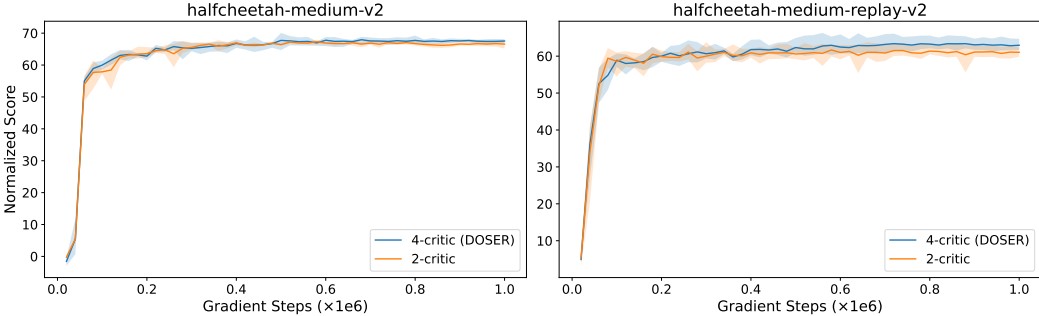

Figure 14: Sensitivity analysis of the number of critic networks.

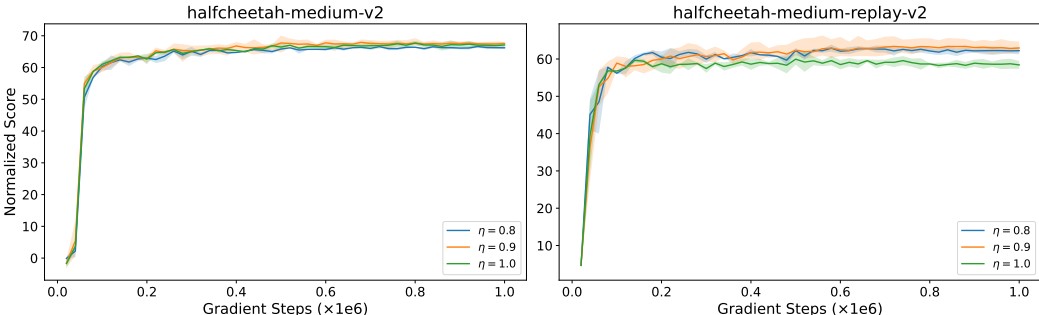

Figure 15: Sensitivity analysis of the compensation target weight $\eta$.

### C.6.3 COMPENSATION TARGET WEIGHT $\eta$

We set the default value of the compensation target weight $\eta$ to 0.9 in all main experiments. This hyperparameter controls the weight of the target Q-value of beneficial OOD actions, a smaller $\eta$ reduces the extent of compensation and makes DOSER more conservative. To evaluate the sensitivity of DOSER to this hyperparameter, we conduct an ablation study by varying $\eta \in \{0.8, 1.0\}$ while keeping all other components unchanged.

As shown in Figure 15, DOSER maintains consistently stable performance across this range. However, when setting $\eta = 1.0$, we observe a slight degradation in performance on halfcheetah-medium-replay-v2. This is primarily due to mild value overestimation introduced by fully adopting the target Q-values of beneficial OOD actions without discounting. Overall, the default choice $\eta = 0.9$ effectively mitigates such overestimation while still enabling meaningful policy improvement.

### C.6.4 THE NUMBER OF SAMPLED IN-DISTRIBUTION ACTIONS $N$

In the OOD action classification stage, DOSER estimates the optimal in-distribution action by sampling $N$ candidate actions from the offline dataset and selecting the one with the highest Q-value as the reference. To assess the robustness of DOSER to the chioce of $N$, we conduct experiments on the halfcheetah tasks with $N \in \{5, 10, 20\}$.

The results in Figure 16 indicate that DOSER maintains strong performance across different values of $N$. A larger $N$ provides a more accurate approximation of the optimal ID action but comes with increased computational cost, whereas a small $N$ may introduce randomness in the estimation. Since the optimal ID Q-value is used only to construct an optimistic Q-target that guides beneficial OOD actions toward higher-value regions, DOSER does not rely heavily on the precise accuracy of this estimate. Therefore, we choose $N = 10$ as a reasonable trade-off between computational efficiency and estimation accuracy in our main experiments.

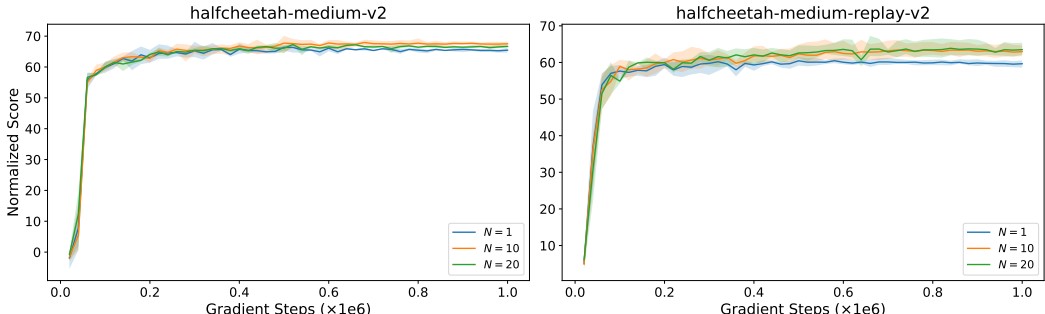

Figure 16: Sensitivity analysis of the number of sampled in-distribution actions $N$.

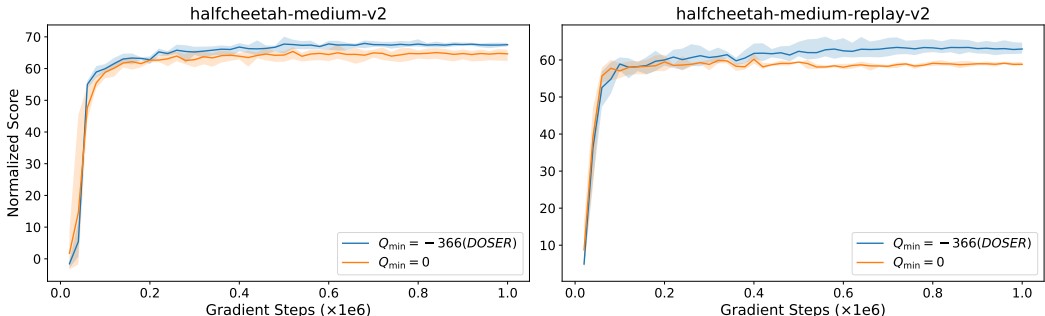

Figure 17: Sensitivity analysis of the value of $Q_{\min}$.

### C.6.5 Q-VALUE LOWER BOUND $Q_{\min}$

DOSER employs a lower bound $Q_{\min}$ when penalizing detrimental OOD actions. In our main experiments, this value is not treated as a tunable hyperparameter. Instead, it is derived directly from the environment dynamics as $Q_{\min} = \frac{R_{\min}}{1-\gamma}$, which corresponds to the standard minimum achievable return under the given discount factor $\gamma$. For the halfcheetah environment, setting $\gamma$ to $0.99$ yields $Q_{\min} = -366$. To further examine the impact of this parameter, we conduct an ablation study in which $Q_{\min}$ is set to $0$ while keeping all other components unchanged. This alternative setting corresponds to a less conservative penalty on potentially detrimental OOD actions. We evaluate this variant on the halfcheetah tasks, with the resulting learning curves reported in Figure 17.

The results show that replacing the original value of $Q_{\min}$ with $0$ leads to performance degradation across both datasets. This outcome can be attributed to the fact that a higher lower bound reduces the penalization applied to detrimental OOD actions, thereby weakening the mechanism designed to mitigate value overestimation. Nevertheless, even with this suboptimal setting, the overall performance remains competitive with existing offline RL baselines.

### C.7 ENSEMBLE-GUIDED GATING MECHANISM

We further introduce an ensemble-guided uncertainty gating mechanism on top of the learned dynamics model, which is designed to prevent unreliable next-state predictions from influencing the classification of OOD actions. We construct an ensemble of $K = 5$ independently initialized dynamics models, each trained on the original offline dataset. For any state-action pair $(\boldsymbol{s}, \boldsymbol{a})$, the ensemble produces multiple next-state predictions $\{\hat{\boldsymbol{s}}'_1, \hat{\boldsymbol{s}}'_2, \ldots, \hat{\boldsymbol{s}}'_K\}$. We calculate the prediction variance across ensemble members as a measure of epistemic uncertainty:

$$\text{Var}(\hat{\boldsymbol{s}}') = \frac{1}{K} \sum_{k=1}^{K} \|\hat{\boldsymbol{s}}'_k - \bar{\boldsymbol{s}}'\|^2, \quad \text{where} \quad \bar{\boldsymbol{s}}' = \frac{1}{K} \sum_{k=1}^{K} \hat{\boldsymbol{s}}'_k \tag{74}$$

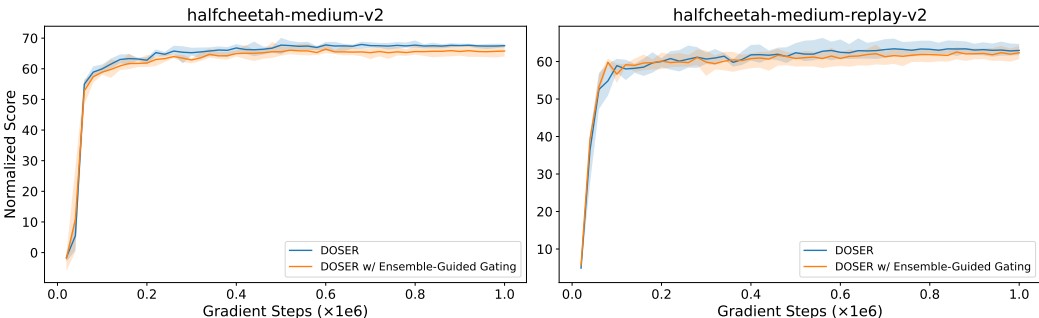

Figure 18: Comparison of DOSER with and without ensemble-guided gating.

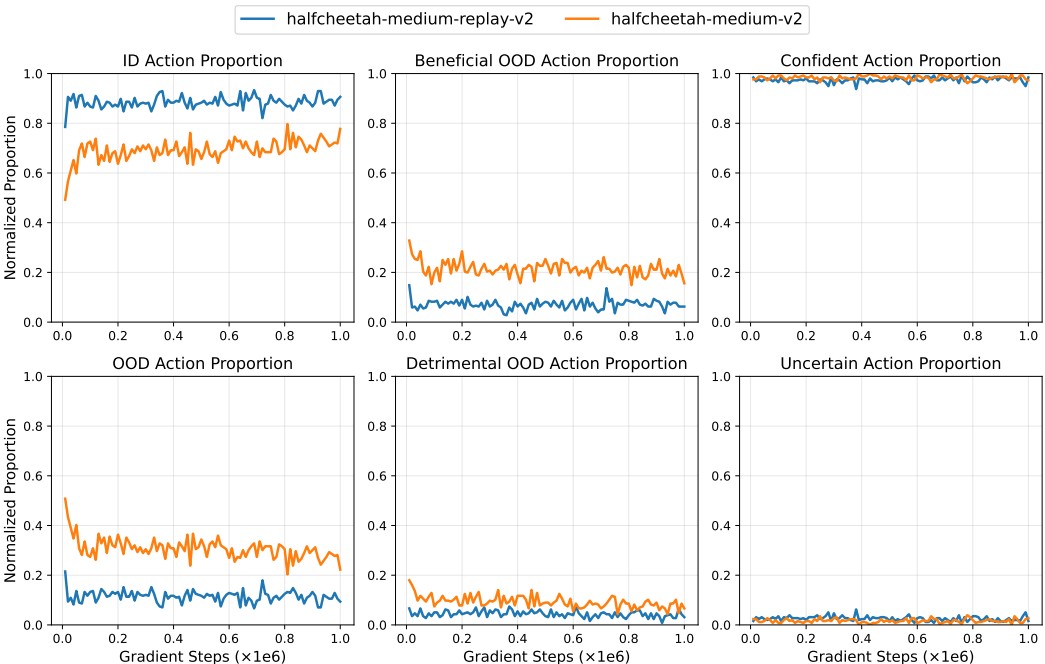

Figure 19: Proportions of different action types with ensemble-guided gating.

To determine whether a predicted next state is reliable, we estimate the empirical distribution of these variances on the offline dataset and use the 99-th percentile as a reliability threshold $\tau_{\mathrm{var}}$. Only when the prediction variance for an OOD action falls below this threshold do we trust the predicted next state and apply the value-based beneficial/detrimental classification; otherwise, the action is conservatively categorized as detrimental.

However, Figure 18 demonstrates that incorporating this ensemble-guided gating mechanism into DOSER brings no noticeable performance improvement in the halfcheetah environments. We further analyze the action type proportions during training (Figure 19), defining confident actions as those with prediction variance below the reliability threshold, and uncertain actions as those filtered out by the gate. The results indicate that fewer than 5% of actions exceed the threshold and are consequently filtered out.

This result again suggests that the pretrained dynamics model already generalizes reasonably well to the moderately OOD regions. It is also possible that the chosen ensemble size of 5 is insufficient to fully capture epistemic uncertainty, and larger or more expressive ensembles might provide stronger gating effects. We leave the exploration of more sophisticated uncertainty quantification methods to future work.

Table 7: Additional performance comparison on Gym-MuJoCo expert and random datasets. We report the mean and standard deviation over 4 seeds for DOSER.

| Dataset | BC | BCQ | BEAR | DT | AWAC | OneStep | TD3+BC | CQL | IQL | DMG | DOSER (Ours) |
|---|---|---|---|---|---|---|---|---|---|---|---|
| halfcheetah-e | **92.9** | 89.9 | **92.7** | 87.7 | 81.7 | 88.2 | **96.7** | 96.3 | 95.0 | 95.9 | 95.4 ± 0.6 |
| hopper-e | **110.9** | **109.0** | 54.6 | 94.2 | **109.5** | 106.9 | 107.8 | 96.5 | 109.4 | 111.5 | 111.6 ± 0.5 |
| walker2d-e | 107.7 | 106.3 | 106.6 | 108.3 | 110.1 | 110.7 | 110.2 | 108.5 | 109.9 | 114.7 | 111.2 ± 0.3 |
| halfcheetah-r | 2.6 | 2.2 | 2.3 | 2.2 | 6.1 | 2.3 | 11.0 | 17.5 | 13.1 | 28.8 | **32.8 ± 1.5** |
| hopper-r | 4.1 | 7.8 | 3.9 | 5.4 | 9.2 | 5.6 | 8.5 | 7.9 | 7.9 | 20.4 | **31.2 ± 0.1** |
| walker2d-r | 1.2 | 4.9 | **12.8** | 2.2 | 0.2 | 6.9 | 1.6 | 5.1 | 5.4 | 4.8 | 3.5 ± 2.3 |
| **Average** | 53.2 | 53.4 | 45.5 | 50.0 | 52.8 | 53.4 | 56.0 | 55.3 | 56.8 | **62.7** | **64.9** |

## C.8 ADDITIONAL EXPERIMENT RESULTS ON D4RL BENCHAMRK

To further validate DOSER's performance across a wider range of dataset qualities, we conduct additional experiments on the Gym-MuJoCo expert and random datasets, as shown in Table 7. Across the expert datasets, DOSER achieves competitive performance relative to prior offline RL methods. More notably, on the random datasets, where the behavior data is highly suboptimal, DOSER exhibits stronger performance, indicating its robustness under poor-quality offline data.

## C.9 LEARNING CURVES

Learning curves on D4RL tasks are provided in Figure 20, Figure 21, and Figure 22. The curves are averaged over 4 random seeds, with the shaded area representing the standard deviation across seeds.

## D THE USE OF LARGE LANGUAGE MODELS (LLMS)

We acknowledge the assistance of GPT-5 in proofreading and polishing the manuscript. The authors bear full responsibility for the content and presentation of this paper.

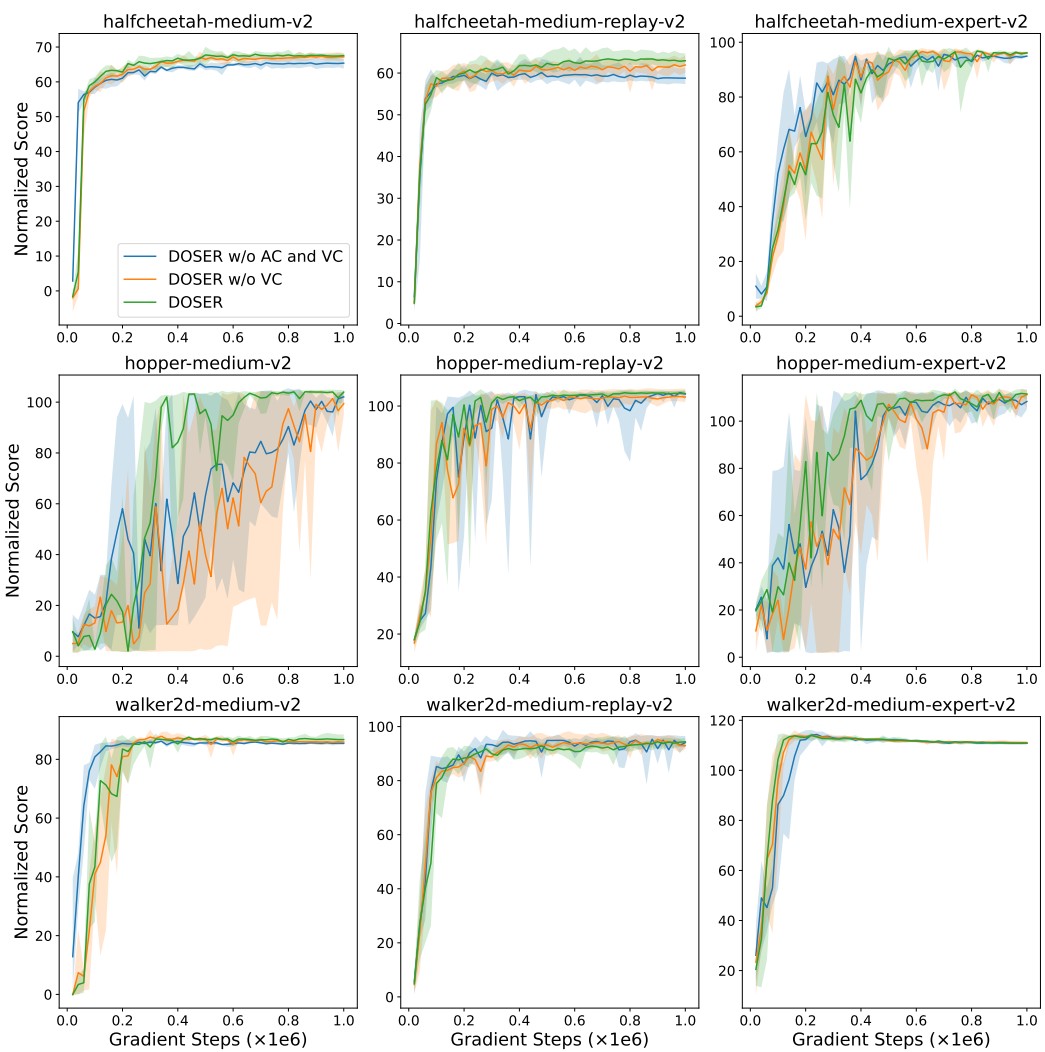

Figure 20: Learning curves of the component ablation study on Gym-MuJoCo tasks.

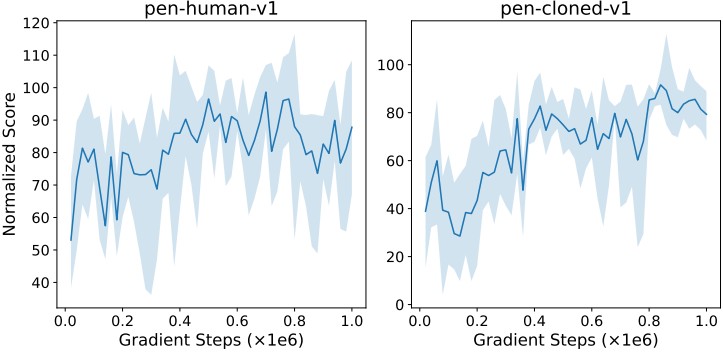

Figure 21: Learning curves on Adroit tasks.

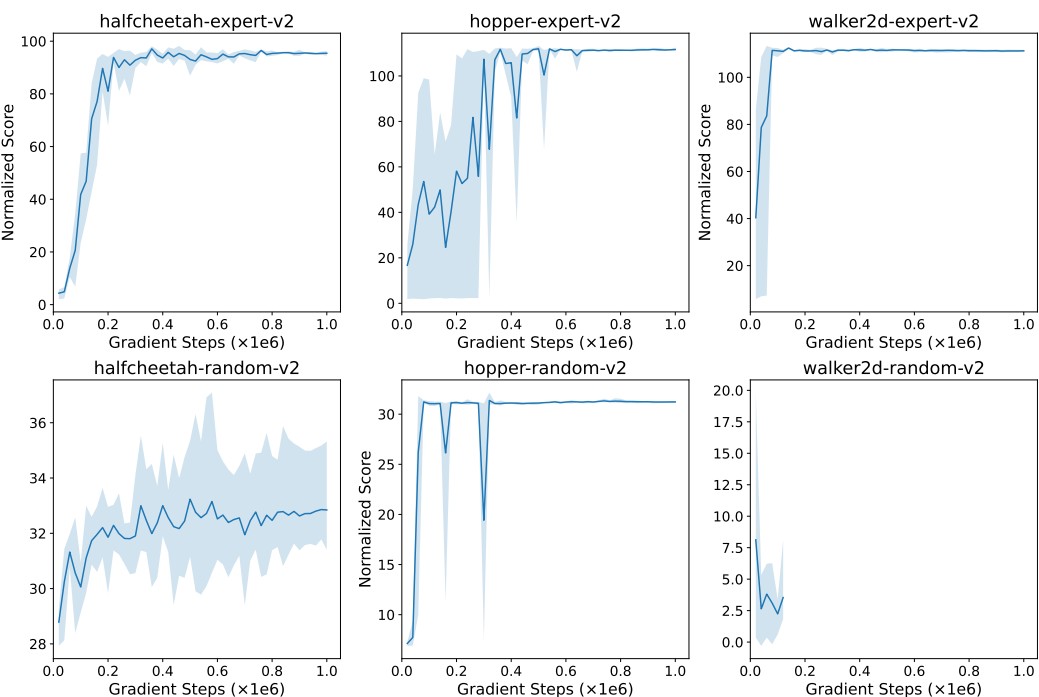

Figure 22: Learning curves on Gym-MuJoCo expert and random tasks.

