# OpenReview forum: "Beyond Penalization: Diffusion-based Out-of-Distribution Detection and Selective Regularization in Offline Reinforcement Learning"
_ICLR.cc/2026/Conference — ICLR 2026 Poster_

### Official Review · Reviewer_9PZb · 2025-10-28

**Soundness:** 3
**Presentation:** 3
**Contribution:** 2
**Rating:** 4
**Confidence:** 4

**Summary:**

The proposed DOSER framework addresses limitations in offline reinforcement learning by using diffusion models to detect out-of-distribution (OOD) actions based on reconstruction error. It then selectively regularizes these actions, suppressing detrimental ones while encouraging those deemed beneficial for exploration.

**Strengths:**

1. The paper is clearly written and well-structured, making its central contributions easy to understand.
2. The proposed method achieves competitive performance on Gym domain.

**Weaknesses:**

1. The method's design is notably complex, incorporating multiple auxiliary models beyond the standard actor-critic setup (e.g., diffusion model for state, diffusion model for action, ensembled dynamics models, implicit value model, implicit target value model, Two extra critic networks). This design challenges the method's practicality and scalability for real-world applications.
2. The method introduces a large set of algorithm-specific hyperparameters ($ \tau_s, \tau_a, \tau, \beta, \lambda, \eta, \delta_{\mathrm{V}}, N, Q_{\min} $). The sensitivity and impact of most of these parameters on the final performance are not adequately analyzed, leaving concerns about the robustness and reproducibility of the results. For instance, the effect of perturbations in \( Q_{\min} \) remains unclear.
3. While the core idea is intuitive, the empirical performance gains appear limited according to the ablation study, with consistent improvements observed on only two tasks (halfCheetah-m and halfCheetah-mr). The proposed method is quite similar to SVR. By comparing their performances, one can draw the conclusion that the benefits are limited for other tasks as well.
4. Could you provide the curves of the selective ratio for ID, OOD-beneficial, and OOD-detrimental actions throughout the training process?
5. There seems to be a potential flaw in the derivation. The claim that $ \eta Q_{\min} \geq Q_{\min} $ is contradictory given the stated parameter range $ \eta \in [0,1) $.
6. The provided performance bound $ Q_{\min} \leq Q^{\pi} \leq Q_{\in}^{\pi} + \eta \delta_{\mathrm{V}} $ is arguably trivial, as it essentially states that the value is bounded by some minimum and maximum. A non-trivial bound with asymptotic analysis, characterizing the error relative to the optimal policy, is needed for a meaningful theoretical contribution.
7. The use of four critic networks, compared to the two commonly used in general methods, may confer an unfair advantage. An ablation study demonstrating performance with only two critics is necessary to ensure an equitable comparison and to isolate the contribution of the core algorithmic innovations.
8. The evaluation would be more comprehensive with the inclusion of the AntMaze domain.

**Questions:**

Please see Weakness

---

> ### Author Response · Authors · 2025-11-24
> **Author Response to Reviewer 9PZb (1/2)**
>
> Dear Reviewer 9PZb:
> We sincerely thank the reviewer for the constructive comments and the appreciation of our work.
> Our response to your concerns/questions:
>
> ### Q1: Concern about the method's complexity.
> ---
> A1: We appreciate the feedback. We acknowledge that DOSER incorporates auxiliary models, but each component serves a critical role in achieving robust OOD handling.
>
> The diffusion models provide reliable reconstruction error signals for OOD detection, while the dynamics model enables OOD classification.
> Importantly, **DOSER uses only a single lightweight dynamics model, not an ensemble.**
> All auxiliary models are pretrained once and kept frozen throughout policy optimization, requiring approximately **2 hours for each diffusion model and under 5 minutes for the dynamics model** on a single RTX 3090 GPU.
>
> Regarding the implicit value networks, we would like to clarify that they are integrated into the critic module and trained jointly with the Q networks, following a design commonly adopted in offline RL[1].
>
> We also conducted an ablation on critic numbers (**See Figure 14 in Appendix C.6.2**), the results show that DOSER retains strong performance even when removing the two extra critics.
>
> Finally, we note that auxiliary models are widely adopted in offline RL.
> For instance, Diffusion Policy with Q-Ensembles[2] uses an ensemble of up to 64 critics.
> In comparison, DOSER offers a balanced trade-off between model complexity and performance.
>
>
> ### Q2: A large set of algorithm-specific hyperparameters.
> ---
> A2: Thank you for raising this important concern.
> As detailed in **Section 4.4 (Figure 5)**, we have thoroughly evaluated the sensitivity of key hyperparameters, including the OOD thresholds ($\tau\_s$, $\tau\_a$), penalty coefficient ($\beta$), and compensation coefficient ($\lambda$).
> The variation in optimal settings across datasets is a natural adaptation to the underlying distribution, as different data coverage requires different trade-offs between conservatism and optimism.
>
> The expectile parameter $\tau$ follows the conventional setup in IQL-style algorithms.
> For the compensation weight $\eta$ and the number of sampled actions $N$, we have added sensitivity analyses in the revision (**see Figure 15 in Appendix C.6.3 and Figure 16 in Appendix C.6.4**).
> Results show that DOSER performs robustly across a wide range of values.
>
> We would like to clarify that $\delta\_V$ is **not a tunable hyperparameter**, but an adaptive value compensation term representing the value difference between two predicted next states.
>
> Regarding $Q\_{\min}$, this is also **not a tunable parameter**, but a theoretical lower bound derived from the MDP, $Q\_{\min} = \dfrac{R\_{\min}}{1 - \gamma}$.
> We also include sensitivity analysis on $Q\_{\min}$ in **Figure 17 (Appendix C.6.5)**.
>
>
> ### Q3: Concern about empirical performance gains.
> ---
> A3: The performance improvement of DOSER is closely related to the dataset characteristics.
> On the halfcheetah-mr and halfcheetah-m datasets, which contain a broad mixture of suboptimal actions, the ability to identify and exploit beneficial OOD actions provides significant value.
> In contrast, for near-optimal datasets such as medium-expert, the room for further improvement through extrapolation is inherently limited.
> This observation aligns with our **action type proportion analysis in Figure 11 (Appendix C.4)**, where the ratio of beneficial to detrimental OOD actions is naturally lower when the dataset is near-optimal.
>
> As shown in **Figure 3 and Figure 4 (Section 4.1)**, diffusion models provide more accurate OOD detection compared to CVAE-based methods like SVR.
> Building on this advantage, DOSER further incorporates selective value regularization, enabling beneficial exploration beyond the dataset support.
>
> Beyond performance metrics, our primary contribution lies in establishing a diffusion-based framework for explicit OOD handling in offline RL.
> We hope that DOSER provides insights and a foundation for future research in this direction.
>
>
> ### Q4: The curves of the selective ratio for ID, OOD-beneficial, and OOD-detrimental actions throughout the training process.
> ---
> A4: Please refer to the **global response Q2**.

---

> ### Author Response · Authors · 2025-11-24
> **Author Response to Reviewer 9PZb (2/2)**
>
> ### Q5: The derivation about $\eta Q\_{\min} \ge Q\_{\min}$.
> ---
> A5: There might be some misunderstanding.
> We would like to clarify that the derivation is correct given that $Q\_{\min}$ is a large **negative** number, as $Q_{\min} = \dfrac{R_{\min}}{1 - \gamma}$ is the theoretical lower bound determined by the MDP.
>
> To prevent confusion, we have explicitly stated in the derivation that $Q\_{\min}<0$, and we have updated **Table 4 in Appendix B.3** to include the actual $Q\_{\min}$ values for all environments.
>
>
> ### Q6: Lack of non-trivial performance bound with asymptotic error analysis.
> ---
> A6: Please refer to the **global response Q4**.
>
>
> ### Q7: Ablation study on the number of critic networks.
> ---
> A7: Thank you for the suggestion.
> We have conducted the ablation study using only two critics, with results detailed in **Appendix C.6.2**.
> The results show that DOSER achieves comparable performance with two critics, indicating that our performance gains do not rely on a larger critic ensemble.
>
>
> ### Q8: The evaluation would be more comprehensive with the inclusion of the AntMaze domain.
> ---
> A8: Please refer to the **global response Q3**.
>
>
> ### Reference
> ---
> [1] Chen, T., Wang, Z., & Zhou, M. (2024). Diffusion policies creating a trust region for offline reinforcement learning. Advances in Neural Information Processing Systems, 37, 50098-50125.
>
> [2] Zhang, R., Luo, Z., Sjölund, J., Schön, T., & Mattsson, P. (2024). Entropy-regularized diffusion policy with q-ensembles for offline reinforcement learning. Advances in Neural Information Processing Systems, 37, 98871-98897.

---

### Official Review · Reviewer_AWU7 · 2025-10-29

**Soundness:** 3
**Presentation:** 3
**Contribution:** 3
**Rating:** 6
**Confidence:** 4

**Summary:**

This paper proposes the DOSER method, a novel framework that goes beyond uniform penalization. DOSER uses two diffusion models to learn the behavior policy and state distribution. Meanwhile, it uses a single-step denoising reconstruction error to detach the OOD action.

**Strengths:**

* DOSER further distinguishes between beneficial and detrimental OOD actions by evaluating predicted transitions, selectively suppressing risky actions while encouraging exploration of high-potential ones.
* DOSER establishes stable convergence with bounded value estimates under $\gamma$-contraction guarantees.

**Weaknesses:**

* This method needs to train two diffusion models, which may bring more computational burden.
* This method also trains a dynamic model and multiple samples, which may bring more computational burden.

**Questions:**

* Similar to the motivation in this paper, there are numerous other methods for achieving efficient policy optimization through action discrimination. For instance, from the perspective of policy constraints, the latest offline RL method A2PR[1] employs advantage functions to select actions that guide policy constraints toward optimization. Could you elaborate on the advantages of DOSER? Additionally, could you include experimental comparisons and discussions with A2PR?

* I'm curious about the trend in the proportion of beneficial OOD actions versus detrimental OOD actions during training for different tasks. Does this significantly impact the training results?

* There are too few main experiments. Could we add some navigation tasks, such as Antmate?



Reference:

[1]Liu, T., Li, Y., Lan, Y., Gao, H., Pan, W., & Xu, X.. Adaptive Advantage-Guided Policy Regularization for Offline Reinforcement Learning. In International Conference on Machine Learning (pp. 31406-31424). PMLR.

---

> ### Author Response · Authors · 2025-11-24
> **Author Response to Reviewer AWU7**
>
> Dear Reviewer AWU7:
> We sincerely thank the reviewer for the constructive comments and the appreciation of our work.
> Our response to your concerns/questions:
>
> ### Q1: This method needs to train two diffusion models, which may bring more computational burden.
> ---
> A1: Thanks for rising this concern.
> We would like to state that employing two diffusion models is essential for achieving robust OOD detection and classification, as they provide reconstruction error signals for both actions and states.
> Our ablation study in **Section 4.3** confirms that removing the diffusion model for state leads to clear performance degradation (*DOSER w/o VC* vs. *DOSER w/o AC and VC*), validating the necessity of this approach.
>
> Importantly, both models are pretrained once and remain frozen during policy learning, so they incur no additional iterative cost during RL optimization.
> In practice, the pretraining of each diffusion model requires only about 2 hours on a single NVIDIA RTX 3090 GPU, which is a modest cost considering the performance gains achieved.
>
>
> ### Q2: This method also trains a dynamic model and multiple samples, which may bring more computational burden.
> ---
> A2: In our framework, the dynamics model is as essential as the diffusion models, as it provides the predictive signal for OOD action classification.
> Similar to the diffusion models, the dynamics model is pretrained and fixed throughout policy optimization.
> The entire pretraining stage takes **less than 5 minutes** on a single NVIDIA RTX 3090 GPU, making its computational cost negligible.
>
> Regarding the sampling process, we selected 10 action samples in our main experiments to balance accuracy and efficiency.
> However, **our newly added sensitivity analysis in Appendix C.6.4 reveals that using only a single sample is sufficient to achieve strong performance**.
> This demonstrates that the sampling process remains computationally efficient and does not pose a practical burden when deploying DOSER.
>
>
> ### Q3: Comparison with A2PR.
> ---
> A3: Thank you for suggesting the comparison with A2PR.
> **We have included A2PR as a baseline in our revision and provide a detailed discussion in Section 4.2.**
>
> A2PR is a CVAE-based policy constraint method whose action discrimination mechanism operates entirely within the dataset.
> It selects high-advantage ID actions to guide policy optimization, leaving exploration restricted to the CVAE's learned support.
> As a result, its performance is inherently limited by both the expressiveness of the CVAE and the coverage of the offline dataset.
>
> In contrast, DOSER is a diffusion-based value regularization method with two key advantages:
> - Diffusion models better capture multi-modal distribution than CVAEs, enabling more accurate characterization of the behavior distribution.
>
> - DOSER's selective regularization explicitly targets OOD actions, allowing the policy to explore high-value regions beyond the dataset support.
>
> Empirically, DOSER outperforms A2PR on Mujoco-v2 tasks, particularly on the medium-replay datasets where modeling multi-modality and exploring beneficial OOD actions are critical.
>
>
> ### Q4: The proportion of beneficial OOD actions versus detrimental OOD actions during training for different tasks.
> ---
> A4: Please refer to the **global response Q2**.
>
>
> ### Q5: There are too few main experiments. Could we add some navigation tasks, such as Antmate?
> ---
> A5: Please refer to the **global response Q3**.

---

### Official Review · Reviewer_DqCJ · 2025-10-30

**Soundness:** 2
**Presentation:** 3
**Contribution:** 3
**Rating:** 6
**Confidence:** 4

**Summary:**

This paper tackles a key challenge in offline reinforcement learning: the overestimation of out-of-distribution (OOD) actions. The authors argue that existing methods, which uniformly penalize all OOD actions, are overly pessimistic and suppress potentially "beneficial" explorations.
The paper introduces DOSER (Diffusion-based OOD Detection and SElective Regularization), a novel framework that moves beyond uniform penalization. DOSER's contributions are twofold:
1. Diffusion-based OOD Detection: It employs two diffusion models, trained on in-distribution actions $\hat{\pi}_{\beta}(a|s)$ and states $d(s)$, respectively. The single-step denoising reconstruction error ($\mathcal{E}_a$ and $\mathcal{E}_s$) is used as a high-fidelity metric to detect OOD samples, which the authors argue is superior to VAEs for capturing complex, multi-modal behavior.
2. Selective Regularization: When the learning policy generates an OOD action $a_{ood}$, DOSER does not immediately penalize it. Instead, it uses a learned dynamics model to predict the next state. It then classifies $a_{ood}$ as detrimental and beneficial. While the detrimental actions are heavily penalized in the Q-function loss, the beneficial actions are compensated in the Q-function loss with a value-difference bonus.

The authors provide a theoretical analysis showing the DOSER operator is a $\gamma$-contraction and results in a value estimate that is bounded by the in-sample value plus the beneficial bonus ($\le Q_{In}^{\pi}(s,a_{id}^{*})+\eta\delta_{V}$). Empirically, DOSER achieves state-of-the-art (SOTA) performance on D4RL benchmarks, particularly on suboptimal "medium" and "medium-replay" datasets.

**Strengths:**

1. **Conceptual Novelty**: The primary strength is the original idea of "selective regularization." By classifying OOD actions into beneficial ($\mathcal{A}_{ood}^{+}$) and detrimental ($\mathcal{A}_{ood}^{-}$) sets, the paper moves beyond the field's preoccupation with uniform pessimism. This is a significant and valuable conceptual contribution.
2. **Modern OOD Detection**: The use of diffusion model reconstruction error as an OOD metric is a strong, modern approach. The argument that it is better suited for multi-modal behavior distributions than VAE-based methods is compelling and well-motivated.
3. **Strong Empirical Results**: The algorithm achieves SOTA performance, especially on the challenging and more realistic "medium" and "medium-replay" datasets. This suggests the method is particularly effective at improving upon suboptimal, heterogeneous data.
4. **Clear Ablations**: The ablation study in Table 2 provides clear evidence that both the Action Classification (AC) and Value Compensation (VC) components are critical to the algorithm's success. This validates the paper's core design.

**Weaknesses:**

1. **The Extrapolation Paradox of the Dynamics Model**: The framework's core classifier (Definition 1) relies on a learned dynamics model to predict the consequence of an OOD action. This is a critical flaw. The model is trained on in-distribution data and has no ground-truth supervision for OOD actions. Therefore, its prediction is an extrapolation that is just as likely to be erroneous as the $Q$-function's. The algorithm relies on an unreliable, "hallucinating" component to make its most critical safety decision.
2. **Extreme Hyperparameter Sensitivity**: The algorithm is not a general, robust method but a "brittle" framework that requires massive, task-specific tuning. Appendix Table 4 reveals that key hypermeters (OOD thresholds $\tau_a, \tau_s$, penalty $\beta$, expectile $\tau$) are different for nearly every single environment-dataset pair. For example, $\tau_a$ is 99th percentile for halfcheetah-medium but 80th for halfcheetah-medium-expert. This lack of generality is a severe practical limitation and makes the SOTA claims less impressive, as they are a product of extensive tuning.

**Questions:**

1. **On the Dynamics Model's Reliability**: Could the authors comment on the "Extrapolation Paradox"? How can the algorithm trust the predicted next state $s'$ (and its resulting $V(s')$ and $\mathcal{E}(s')$) when it comes from an OOD action $a_{ood}$ that the dynamics model $p_{\psi}$ has never seen? Have the authors considered using uncertainty estimates from $p_{\psi}$ (e.g., via an ensemble) to gate this classification?
2. **On Practical Hyperparameter Setting**: The reliance on per-task OOD thresholds (80th vs 99th percentile) shown in Table 4 is a major concern. How would a practitioner set $\tau_a$ and $\tau_s$ for a new, unseen offline dataset without a massive tuning budget? Is there a more robust or adaptive way to set this threshold?

---

> ### Author Response · Authors · 2025-11-24
> **Author Response to Reviewer DqCJ**
>
> Dear Reviewer DqCJ:
> We sincerely thank the reviewer for the constructive comments and the appreciation of our work.
> Our response to your concerns/questions:
>
> ### Q1: The extrapolation paradox of the dynamics model.
> ---
> A1: Please refer to the **global response Q1**.
>
> ### Q2: Have the authors considered using uncertainty estimates from $p\_\psi$ (e.g., via an ensemble) to gate this classification?
> ---
> A2: This is an excellent suggestion.
> Indeed, we can adopt an ensemble of dynamics models as a gating mechanism, and we have incorporated this into the revision.
> **Please refer to Appendix C.7.**
>
> We compute the prediction variance across ensemble members and use the 99-th percentile of ID variance as a reliability threshold.
> Only when the variance for an OOD action falls below this threshold do we trust the predicted next state for beneficial/detrimental classification; otherwise, the action is conservatively categorized as detrimental.
>
> However, we empirically find that incorporating this ensemble does not lead to a noticeable performance improvement.
> Our analysis shows that very few actions (less than 5%) exceed the reliability threshold and are filtered out by the ensemble.
>
> This result indicates that the pretrained dynamics model already generalizes reasonably well to the relevant OOD region.
> It is also possible that our current ensemble size of 5 is insufficient to fully capture epistemic uncertainty.
> We consider exploring larger or more expressive ensembles as a promising direction for future work, and we appreciate the reviewer for highlighting this avenue.
>
>
> ### Q3: Extreme hyperparameter sensitivity.
> ---
> A3: Thank you for raising this important point.
> We would like to clarify that the different hyperparameter settings reported in Appendix Table 4 reflect a principled adaptation to dataset characteristics, rather than brittle sensitivity in the algorithm.
>
> In practice, for all MuJoCo tasks, DOSER uses **only two distinct configurations**.
>
> For **narrow-support datasets** (e.g., medium-expert, expert), which contain high-quality but limited behaviors, we adopt a **more conservative** configuration: a lower OOD threshold (80th percentile), a larger penalty coefficient ($\beta=0.05$), and a lower expectile ($\tau=0.7$) to effectively prevent overestimation.
>
> For **broad-support datasets** (e.g., medium, medium-replay, random) with diverse but suboptimal trajectories, we employ a **more optimistic** configuration: a higher OOD threshold (99th percentile), a smaller penalty coefficient ($\beta=0.001$), and a higher expectile ($\tau=0.9$) to avoid excessive pessimism and enable beneficial extrapolation.
>
> In addition, our **sensitivity analysis in Section 4.4** confirms that performance remains stable within each category, and the key is selecting the appropriate configuration type, rather than task-level tuning.
>
>
> ### Q4: How would a practitioner set $\tau\_a$ and $\tau\_s$ for a new, unseen offline dataset without a massive tuning budget? Is there a more robust or adaptive way to set this threshold?
> ---
> A4: When applying our method to a new offline dataset, practitioners can infer the dataset’s coverage by analyzing the shape of the reconstruction error distribution generated by the diffusion model.
>
> In particular, statistics such as **the ratio of the Interquartile Range to the Median (IQR/Median)** serves as an effective indicator of how concentrated or dispersed the dataset support is.
>
> For example, the IQR/Median ratios for halfcheetah-medium-replay and halfcheetah-medium-expert are $0.69$ and $0.60$ respectively, indicating that medium-replay has a more dispersed error distribution.
> This observation aligns with our experimental setting, where we use a higher percentile threshold for medium-replay and a lower one for medium-expert.
> Such a strategy can provide a way for effective threshold selecting.

---

### Official Review · Reviewer_zxuo · 2025-10-31

**Soundness:** 3
**Presentation:** 4
**Contribution:** 3
**Rating:** 4
**Confidence:** 3

**Summary:**

This paper proposes DOSER (Diffusion-based OOD Detection and Selective Regularization), an offline reinforcement learning (RL) framework that aims to mitigate overestimation caused by out-of-distribution (OOD) actions.
The method trains two diffusion models to model the behavior policy and the state distribution, respectively, and uses the single-step denoising reconstruction error as an OOD indicator.
Then, during policy optimization, DOSER further classifies OOD actions into “beneficial” and “detrimental” using a learned dynamics model and value comparison, selectively penalizing or rewarding them.
The authors provide theoretical analysis (γ-contraction and bounded convergence) and empirical results on D4RL MuJoCo benchmarks, claiming consistent improvements over existing VAE-based and diffusion-based offline RL baselines.

**Strengths:**

Interesting integration of diffusion models into OOD detection. The idea of using denoising reconstruction error as a likelihood-free OOD signal is conceptually appealing and avoids Gaussian assumptions used in VAE-based behavior modeling.

Selective regularization is an intuitive extension to uniform penalization — encouraging “good” OOD exploration rather than blindly suppressing all deviations.

Clarity and structure: the paper is generally well written and easy to follow, with clear diagrams (e.g., Fig. 1–2) and algorithmic pseudocode.

Comprehensive empirical comparison: the authors evaluate against both conventional (CQL, IQL, ACL-QL) and diffusion-based (DQL, SRPO, DTQL, etc.) baselines, and perform ablation and sensitivity studies.

**Weaknesses:**

Questionable reliability of OOD classification.
The core contribution—distinguishing beneficial from detrimental OOD actions—relies heavily on a learned dynamics model and single-step value predictions.
In offline RL, both are notoriously unreliable outside the data distribution, making this classification potentially unstable or self-reinforcing. The paper lacks evidence (e.g., accuracy, calibration, or failure cases) showing that such classification can be trusted.

Circular dependency between detection and evaluation.
The method uses diffusion-based reconstruction error to decide OODness and then uses a dynamics model (also trained on the same data) to judge whether the OOD transition is beneficial.
Without ground truth labels or uncertainty calibration, this process risks systematic bias amplification rather than true OOD discrimination.

Limited novelty beyond combining existing ideas.
Diffusion-based behavior modeling (e.g., DQL, IDQL, SRPO) and reconstruction-based OOD detection (e.g., Graham et al., CVPR 2023) are established.
The proposed framework largely combines these with a hand-crafted “bonus/penalty” heuristic; the theoretical part simply restates a standard contraction argument.

Experimental validation does not fully support the claims.
The performance gains on D4RL are moderate and sometimes within variance ranges; ablations do not directly demonstrate that the OOD classification truly identifies beneficial cases.
No visualization, quantitative detection metrics (e.g., AUROC on synthetic OOD test), or robustness analysis are provided to substantiate the detection accuracy.

**Questions:**

How stable is the beneficial vs. detrimental classification in practice? Can the authors report the proportion of actions identified as each type and the accuracy of this classification on a synthetic dataset where ground-truth OOD actions are known?

Does the diffusion reconstruction error correlate quantitatively with true behavior support likelihood (e.g., log-likelihood from a reference model)?

How sensitive is DOSER to errors in the learned dynamics model? Would performance degrade sharply if the model mispredicts next states for unseen regions?

Could the authors show qualitative visualizations (e.g., state-action heatmaps) to verify that “beneficial” OOD actions indeed lead to higher-value regions?

Have the authors tested on harder datasets (Adroit, AntMaze) to demonstrate generalization beyond MuJoCo locomotion?

---

> ### Author Response · Authors · 2025-11-24
> **Author Response to Reviewer zxuo (1/2)**
>
> Dear Reviewer zxuo:
> We sincerely thank the reviewer for the constructive comments and the appreciation of our work.
> Our response to your concerns/questions:
>
> ### Q1: Questionable reliability of OOD classification.
> ---
> A1: Please refer to the **global response Q1**.
>
>
> ### Q2: Circular dependency between detection and evaluation.
> ---
> A2: We thank the reviewer for raising this concern.
>
> We would like to clarify that **DOSER does not suffer from circular dependency or systematic bias amplification**.
> Since both the diffusion models and the dynamics model are pretrained and remain fixed during policy optimization, the detection and evaluation processes are decoupled and do not iteratively influence each other.
>
> Even if the diffusion model occasionally misclassifies an ID action as OOD, the action will almost always be classified as a detrimental OOD, because the beneficial OOD case requires both transitioning to ID region and achieving higher value than the optimal ID action, which is a challenging condition to meet.
> Such misclassified actions are therefore conservatively pushed toward $Q_{\min}$, limiting their influence rather than amplifying errors.
>
> Conversely, if an OOD action is mistakenly classified as ID, it simply bypasses DOSER’s OOD classification and receives neither bonus nor penalty, avoiding a self-reinforcing feedback loop.
>
>
> ### Q3: Limited novelty beyond combining existing ideas.
> ---
> A3: We thank the reviewer for the thoughtful assessment. However, we would like to elaborate further on the novelty of our method:
>
> To the best of our knowledge, existing diffusion-based offline RL methods primarily use diffusion as an improved policy or behavior model, but **none of them targets OOD detection itself, which is a fundamental challenge in offline RL**.
> Our work takes a different perspective by repurposing diffusion models explicitly for OOD handling, and we hope this paradigm will inspire future research into more robust and principled OOD detection methods.
>
> Unlike existing approaches which treat all OOD actions as uniformly harmful, DOSER introduce a discriminative mechanism that **goes beyond binary ID/OOD treatment to distinguish between beneficial and detrimental OOD actions**.
>
> Moreover, the bonus/penalty term is not a simple heuristic but a theoretically grounded instantiation of selective regularization, and **we have further added asymptotic error analyses to make our theoretical part more complete in the revision**.
> Please refer to the **global response Q4**.
>
>
> ### Q4: Experimental validation does not fully support the claims.
> ---
> A4: We really appreciate your insightful suggestion.
> We have strengthened our empirical analysis in the revision to better substantiate our claims.
>
> **Quantitative validation.**
> We construct a synthetic OOD dataset by perturbing in-distribution actions with Gaussian noise at varying scales and report standard OOD detection metrics.
> Our diffusion-based method achieves high AUROC scores (e.g., 0.998 under noise scale 1.0), demonstrating its reliability in OOD detection.
> **Please refer to Table 5 and Figure 9 in Appendix C.1.**
>
> To further verify generalizability beyond D4RL, we additionally evaluate the diffusion-based reconstruction error on common OOD detection benchmarks.
> Our method outperforms or matches existing baselines.
> **Please refer to Table 6 in Appendix C.2.**
>
> **Qualitative visualizations.**
> For the synthetic OOD dataset, we visualize the reconstrution error distributions across noise scales **(Figure 8 in Appendix C.1)**.
>
> We use the learned critic to visualize Q-value distributions of ID actions, beneficial OOD actions, and detrimental OOD actions.
> The results show that beneficial OOD actions consistently receive higher Q-values, while detrimental ones primarily fall in the lower value range.
> **Please refer to Figure 12 in Appendix C.5.**
>
>
> ### Q5: How stable is the beneficial vs. detrimental classification in practice? Can the authors report the proportion of actions identified as each type and the accuracy of this classification on a synthetic dataset where ground-truth OOD actions are known?
> ---
> A5: We report the proportions of each action type during training, please refer to the **global response Q2**.
> The beneficial vs. detrimental classification is stable across tasks, demonstrating that the classifier does not collapse during policy optimization.
>
> We construct a **synthetic OOD dataset** by perturbing actions from the D4RL dataset with controlled Gaussian noise, which provides explicit ID/OOD labels.
> We then report Accuracy, Precision, Recall, F1-Score, and AUROC for the diffusion-based OOD detector.
> Even under small perturbations where ID and OOD heavily overlap, our method maintains high AUROC, and all detection metrics approach near-perfect values as the noise level increases.
> **Please refer to Table 5 and Figure 9 in Appendix C.1.**

---

> ### Author Response · Authors · 2025-11-24
> **Author Response to Reviewer zxuo (2/2)**
>
> ### Q6: Does the diffusion reconstruction error correlate quantitatively with true behavior support likelihood (e.g., log-likelihood from a reference model)?
> ---
> A6: We added quantitative correlation analysis between diffusion reconstruction error and negative log-likelihood.
> **Please refer to Figure 10 in Appendix C.3.**
>
> On a Gaussian mixture with known ground-truth density, the diffusion reconstruction error exhibits a very strong Pearson correlation with the true negative log-likelihood ($\rho = 0.9718$).
>
> We further conduct analysis on the halfcheetah-medium-replay-v2 dataset, with CVAE as a reference model to approximate the behavior support likelihood.
> The diffusion reconstruction error again exhibits a strong positive correlation with the CVAE-based NLL ($\rho=0.7848$), despite the CVAE’s limitations in modeling multi-modal distributions.
>
> Therefore, we have reason to believe that the diffusion reconstruction error serves as a theoretically grounded metric for OOD detection.
>
>
> ### Q7: How sensitive is DOSER to errors in the learned dynamics model? Would performance degrade sharply if the model mispredicts next states for unseen regions?
> ---
> A7: **We have added a sensitivity analysis of the dynamics model error in Appendix C.6.1.**
> By using early-stage checkpoints of the dynamics model, we observe that performance indeed degrades as the model becomes less accurate.
> Crucially, this degradation is not sharp or catastrophic, demonstrating that DOSER is reasonably robust to moderate prediction inaccuracies in the dynamics model.
>
> At the same time, the comparison highlights the effectiveness of a fully trained dynamics model used in DOSER, suggesting that it possesses reasonable generalization capability towards moderately OOD regions.
>
>
> ### Q8: Could the authors show qualitative visualizations (e.g., state-action heatmaps) to verify that “beneficial” OOD actions indeed lead to higher-value regions?
> ---
> A8: Thank you for this valuable suggestion.
> Since the true higher-value regions of the environment are not directly measurable, we use the learned critic as a value estimator and visualize the Q-value distributions of ID actions, beneficial OOD actions, and detrimental OOD actions.
> **Please refer to Figure 12 in Appendix C.5.**
>
> We observe that beneficial OOD actions consistently exhibit higher estimated Q-values, whereas the Q-value distribution of detrimental OOD actions are primarily near the lower tail.
> This provides qualitative support that the classifier successfully identifies OOD actions that lead toward higher-value regions.
>
>
> ### Q9: Have the authors tested on harder datasets (Adroit, AntMaze) to demonstrate generalization beyond MuJoCo locomotion?
> ---
> A9: Please refer to the **global response Q3**.

---

### Author Response · Authors · 2025-11-24
**Global response by the authors (1/2)**

We sincerely thank the reviewers, ACs and PCs for ensuring high-quality review of the paper.
We find all reviews constructive and helpful for making our paper stronger.

Here we summarize some key/common questions raised and provide our general response as follows:

### Q1: Reliability of the learned dynamics model and state value estimation for OOD action classification.
---
A1:
We sincerely thank the reviewers for raising this important concern.

(1) Effectiveness of the learned dynamics model.

As shown in **component ablation in Section 4.3**, incorporating the learned dynamics model indeed provides a reliable signal for OOD action classification and consistently achieves better performance.

(2) Preventing extrapolation error of the model.

It is important to note that after predicting the next state, we apply state-level OOD detection to ensure that **value estimation is performed only on in-distribution states**, thereby mitigate the extrapolation error on OOD states.

(3) Sensitivity analysis on model accuracy.

We further conduct a new sensitivity analysis by replacing the fully trained dynamics model with early-stage checkpoints that exhibit substantially higher prediction errors (**See Figure 13 in Appendix C.6.1**).
We observe that using poorly trained models indeed degrades the overall performance.
This confirms that the accuracy of the dynamics model matters; at the same time, it also verifies that **the fully trained dynamics model does possess meaningful generalization ability beyond the dataset support**.

Thanks to the reviewer DqCJ's excellent suggestion, we also attempt to introduce a model ensemble to gate the classification, using the prediction variance to select reliable transitions for the subsequent OOD action classification.
We find that only a very small number of actions are filtered out by this gate, which further validates the reliability of the pretrained dynamics model.
**Please refer to Appendix C.7 for detailed implementation.**


### Q2: The proportion of ID, beneficial OOD, and detrimental OOD actions during the training process on different tasks.
---
A2: We thank the reviewer for raising this valuable question.
We have analyzed the proportion of different action types during training, as shown in **Figure 11 (Appendix C.4).**

Across all tasks, **the proportion of ID actions increases as training progresses**, indicating that the learned policy gradually aligns with the behavior support.
However, the relative balance between different actions varies across datasets.
Medium-replay exhibits the highest ID ratio, followed by medium, whereas medium-expert shows the lowest.
This is consistent with the underlying data coverage, as **a narrower support naturally leads to more OOD actions**.

Among OOD actions, **the proportion of beneficial OOD actions is substantially larger than that of detrimental ones** on the **medium-replay and medium** datasets.
In contrast, on the **medium-expert** dataset, **these two types of OOD actions have an almost equal proportion**.
This pattern reflects the quality of the offline dataset, since the medium-expert contains more near-optimal behaviors, the potential performance gain from extrapolation is limited.


### Q3: Experiments on harder datasets beyond MuJoCo locomotion tasks.
---
A3: We thank the reviewer for this valuable suggestion.
Our implementation is built upon the SVR[1] codebase, which unfortunately is unable to learn meaningful policies in the Antmaze domain due to inherent limitations in its pipeline.

To strengthen our empirical evaluation beyond Mujoco locomotion tasks, we have added experiments on the **Adroit manipulation tasks** in the revision.
As shown in **Table 1**, DOSER achieves state-of-the-art performance on the two pen tasks, outperforming most existing offline RL baselines.
The corresponding learning curves are provided in **Figure 21 (Appedix C.9)**.

Additionally, we have expanded our evaluation to include **MuJoCo expert and random datasets**, the full results are presented in **Table 7 (Appendix C.8)**.

Extending DOSER to navigation domains such as Antmaze is an important next step, and we regard this as a promising direction for future work.

---

### Author Response · Authors · 2025-11-24
**Global response by the authors (2/2)**

### Q4: Concerns about the theoretical part. / Lack of non-trivial performance bound with asymptotic error analysis.
---
A4: We thank the reviewer for this valuable feedback.
In the revision, we have strengthened the theoretical foundation of DOSER by introducing **two new theorems (Theorem 3 and Theorem 4) in Appendix A.2**, which establish a critic deviation bound and an asymptotic performance guarantee relative to the optimal policy.
Below we summarize the key results and proof ideas.

**Theorem 3 (Bounded critic deviation)**

Let $\pi\_{\mathrm{ref}}$ denote the reference policy obtained with the true environment dynamics $P$ and without OOD detection error, with $Q\^{\pi\_{\mathrm{ref}}}$ being its corresponding action-value function.
Let $\widehat{\pi}$ denote the learned policy of DOSER under a dynamics model approximation error $\varepsilon\_{\mathrm{dyn}}$ and an OOD detection misclassification probability $\varepsilon\_{\mathrm{det}}$.
Then, the deviation of the learned critic $\widehat{Q}$ from $Q\^{\pi\_{\mathrm{ref}}}$ is bounded as follows:
$$
||\widehat{Q} - Q\^{\pi_{\mathrm{ref}}}||\_\infty \le \frac{\gamma}{1 - \gamma} \left( Q\_{\max}\left( C\_1 \varepsilon\_{\mathrm{dyn}} + C\_2 \varepsilon\_{\mathrm{det}} \right) + \eta \delta\_V \right),
$$
where $Q\_{\max} = \dfrac{R\_{\max}}{1 - \gamma}$, and $C\_1$, $C\_2$ are constants that capture the sensitivity of the policy optimization process to dynamics and detection errors respectively.

**Proof sketch.**
We compare the DOSER Bellman operator with the reference Bellman operator and decompose the error into three sources: (i) dynamics model approximation error $\varepsilon\_{\text{dyn}}$,
(ii) OOD detector misclassification error $\varepsilon\_{\text{det}}$, and (iii) value compensation term $\eta\delta\_V$.
The final bound is derived using the $\gamma$-contractive property established in Theorem 1.

**Theorem 4 (Performance gap of DOSER)**

Let $\widehat{\pi}$ be the policy learned by DOSER through iterative application of $\mathcal{T}\_{\mathrm{DOSER}}$, and let $\pi\^\*$ denote the optimal policy.
Suppose $\delta\_f$ represents the function approximation error.
Then the performance gap between $\pi\^\*$ and $\widehat{\pi}$ satisfies
$$
\left| J(\pi\^\*) - J(\widehat{\pi}) \right| \leq \delta\_f + \frac{C L\_P R\_{\max}}{1 - \gamma} \left( C\_1 \varepsilon\_{\mathrm{dyn}} + C\_2 \varepsilon\_{\mathrm{det}} \right),
$$
where $C\_1$, $C\_2$ are positive constants, and $L\_P$ is the Lipschitz constant of the environment dynamics.

**Proof sketch.**
We decompose the performance gap into the function approximation error between $\pi\^\*$ and $\pi\_{\mathrm{ref}}$, and the policy deviation between $\pi\_{\mathrm{ref}}$ and $\widehat{\pi}$.
We then bound the latter using the errors from Theorem 3 and analyze the difference in state occupancies to link it back to the final performance difference.

**For complete theoretical derivations, please refer to Appendix A.3.3 and A.3.4.**

### Reference
---
[1] Mao, Y., Zhang, H., Chen, C., Xu, Y., & Ji, X. (2023). Supported value regularization for offline reinforcement learning. Advances in Neural Information Processing Systems, 36, 40587-40609.

---

### Meta-Review · Area_Chair_z6rq · 2026-01-06

**Summary:**

This paper proposes DOSER, an offline RL framework that uses diffusion-model single-step denoising reconstruction error for OOD detection (for both actions and states), then selectively regularizes OOD actions by classifying them as “beneficial” vs “detrimental” using a learned dynamics model and value comparisons, applying bonuses/penalties accordingly. Reviewers liked the overall clarity, strong empirical suite and the idea of selective (non-uniform) OOD treatment, but the key decision concerns were whether the beneficial/detrimental classifier is reliable given model-based extrapolation on OOD regions, whether the method is overly complex and hyperparameter-heavy (potentially brittle), whether the novelty and theory go beyond standard contraction-style arguments, and whether experiments/analyses sufficiently validate the OOD detector and the classification mechanism (including harder domains like AntMaze/Adroit).

**Reviewer Concerns:**

The rebuttal addressed many of the core requests: the authors added quantitative OOD detection validation (synthetic Gaussian-noise OOD with AUROC/accuracy metrics), correlation analysis between reconstruction error and (approximate/true) NLL, dynamics-model sensitivity studies, and visualizations comparing Q-value distributions of ID vs beneficial/detrimental OOD actions; they also reported the proportions of action types during training and added an ensemble-based uncertainty gate for the dynamics classifier (showing it filters few samples). They included A2PR as a baseline and ran an ablation reducing critics to two to address “unfair advantage,” and clarified compute costs by noting auxiliary models are pretrained and frozen (with stated pretraining times) plus a sampling sensitivity study showing few samples suffice. Remaining concerns are mainly about real-world robustness and generality: reliance on learned dynamics/value for OOD classification is still a conceptual risk (even if mitigated empirically), the method remains fairly complex with many knobs, and some reviewers still want stronger evidence on harder benchmarks like AntMaze/Adroit (the response points to a global reply, but details aren’t included in the text provided here).

**Reviewer Scores:**

zxuo likely 4→5 (the added AUROC/correlation/sensitivity/visualizations directly address the main skepticism). DqCJ likely 6→6 (already positive; uncertainty-gating and threshold-setting heuristics help but may not change score). AWU7 likely 6→6 (A2PR comparison + compute clarifications address questions; likely stable). 9PZb likely 4→5 (two-critic ablation, added hyperparameter sensitivities, and clarified “complexity” concerns reduce the main objections, though complexity may still temper enthusiasm).

---

### Decision · Program_Chairs · 2026-01-26

Accept (Poster)